# What Variables Affect Out-of-Distribution Generalization in Pretrained Models?

**Md Yousuf Harun**[1,*]     **Kyungbok Lee**[2,*]     **Jhair Gallardo**[1]
**Giri Krishnan**[3]     **Christopher Kanan**[2]
[1]Rochester Institute of Technology     [2]University of Rochester     [3]Georgia Tech

## Abstract

Embeddings produced by pre-trained deep neural networks (DNNs) are widely used; however, their efficacy for downstream tasks can vary widely. We study the factors influencing transferability and out-of-distribution (OOD) generalization of pre-trained DNN embeddings through the lens of the tunnel effect hypothesis, which is closely related to intermediate neural collapse. This hypothesis suggests that deeper DNN layers compress representations and hinder OOD generalization. Contrary to earlier work, our experiments show this is not a universal phenomenon. We comprehensively investigate the impact of DNN architecture, training data, image resolution, and augmentations on transferability. We identify that training with high-resolution datasets containing many classes greatly reduces representation compression and improves transferability. Our results emphasize the danger of generalizing findings from toy datasets to broader contexts.

## 1  Introduction

Understanding deep neural network (DNN) representations has been a central focus of the deep learning community. It is generally accepted that a DNN's initial layers learn transferable universal features, with deeper layers being task-specific [1–8]. At NeurIPS-2023, [9] challenged this view with evidence for the tunnel effect hypothesis:

> **The Tunnel Effect Hypothesis:** An overparameterized $N$-layer DNN forms two distinct groups:
>
> 1. *The extractor* consists of the first $K$ layers, creating linearly separable representations.
> 2. *The tunnel* comprises the remaining $N - K$ layers, compressing representations and hindering OOD generalization.

To test the tunnel effect hypothesis, they trained models on an in-distribution (ID) dataset and compared linear probes trained and evaluated on either ID or OOD datasets for embeddings produced by each DNN layer. They showed that ID accuracy increased monotonically, whereas OOD accuracy rapidly decreased after the extractor (Fig. 1). The likely cause of the tunnel effect is intermediate neural collapse [10]. Both [10] and [9] showed that collapsed/tunnel layers could be static without needing learning. If these results are universal, they suggest universal visual features learned in early layers and using embeddings from the penultimate layers of pre-trained DNNs, need to be rethought. However, both [9] and [10] limited their experiments to datasets with low-resolution images and relatively few categories (CIFAR-10, MNIST). Given the widespread use of embeddings from pre-trained DNNs for downstream OOD tasks, we aim to assess the universality of the tunnel effect and the variables that influence its strength[†].

---

[*]Equal contribution. Corresponding author: Md Yousuf Harun (mh1023@rit.edu)

[†]Project website and code: `https://yousuf907.github.io/oodg`

38th Conference on Neural Information Processing Systems (NeurIPS 2024).

If the tunnel effect is stronger in models trained on toy datasets like CIFAR-10 but diminishes for large-scale datasets, this may explain why many algorithms evaluated only on toy datasets are ineffective on larger datasets such as ImageNet. This disparity has persisted across problem settings, including open set classification [11], active learning [12], OOD detection [13], uncertainty quantification [14], dataset distillation [15], and continual learning [16–18].

**Our paper makes the following contributions:**

1. We define metrics to measure the *strength* of the tunnel effect and use a SHAP-based analysis to assess each variable's impact, e.g., image resolution, number of semantic classes, and DNN architecture. Using 64 pre-trained ID backbones and 8,604 linear probes, we identify conditions that exacerbate, reduce, and eliminate the tunnel effect.
2. Using our metrics, we find that widely used ImageNet-1K pre-trained CNN and ViT backbones do not exhibit tunnels, except for ResNet-50.
3. In contrast to [9], we find that the tunnel strongly impacts forgetting in continual learning. This suggests the generality of many continual learning systems depends on tunnel strength, which is heavily influenced by architectural and training dataset choices.
4. We establish a link between impaired OOD generalization and the characteristics of widely used toy datasets, with both resolution and a small number of classes exacerbating the tunnel effect.
5. We propose a revised tunnel effect hypothesis, in which the tunnel's *strength* is influenced by training data diversity.

## 2 Related Work

### 2.1 The Tunnel Effect

Strong evidence for the tunnel effect was given in [9], but their experiments are limited. First, they only study MLPs and CNNs, whereas ViT models are now widely used. Second, their experiments only use $32 \times 32$ images, and we hypothesize that higher resolution images could mitigate the tunnel effect by promoting learning hierarchical representations. Third, they do not control for the impact of data augmentation, where data augmentation is known to improve OOD generalization [19–24]. Lastly, they define tunnel as starting at the layer where a linear probe on the ID dataset achieves at least 95% of the final ID accuracy, ignoring OOD generalization. This is problematic since OOD generalization is central to their tunnel effect definition. Here, we measure *tunnel effect strength* using OOD performance.

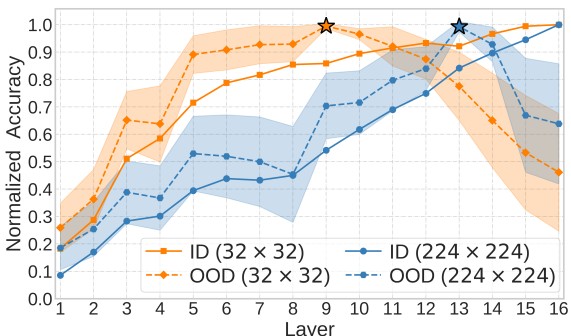

Figure 1: **The tunnel effect.** The tunnel impedes OOD generalization, which we study using linear probes trained on ID and OOD datasets for each layer. In this example, identical VGGm-17 architectures are trained on identical ID datasets, where only the resolution is changed. Probe accuracy on OOD datasets decreases once the tunnel is reached (denoted by ☆), where the model trained on low-resolution ($32 \times 32$) images creates a longer tunnel (layers 9-16) than the one (layers 13-16) trained on higher-resolution ($224 \times 224$) images. The Y-axis shows the normalized accuracy. The OOD curve is the average of 8 OOD datasets (Sec. 3.3), with the standard deviation denoted with shading.

### 2.2 Learning Embeddings that Generalize

Pretrained DNNs are widely used to produce embeddings for downstream tasks, e.g., DNNs trained on ImageNet generalize effectively to many computer vision tasks [25–27]. Several studies [28–31] observe that ImageNet ID accuracy is highly predictive of OOD accuracy, while others [22, 32–34] find that observation does not always hold. Another example is CLIP [35], which has better OOD generalization due to larger and more diverse training data [36]. We disentangle the role of data quantity versus the level of semantic variability.

Many works suggest that representations learned in earlier layers are more universal across image datasets whereas later layers are more task-specific [1–8, 32]. This observation has guided the development of many transfer and continual learning methods. We revisit this phenomenon by studying the impact of DNN architecture on OOD generalization.

Previous studies have focused on *independently* analyzing variables that may impact OOD generalization [3, 4, 9, 22, 32, 36–40]. However, there is still a significant gap in our current understanding regarding each variable's *relative* importance. Our study bridges this gap.

## 3 Methods

### 3.1 Measuring the Tunnel Effect

Following [9], we use linear probes for our tunnel effect analysis. Linear probes are widely used to evaluate the transferability of learned embeddings to OOD datasets [9, 41–52]. After supervised training of a DNN on an ID dataset, we train ID and OOD linear probes on embeddings produced by each layer. Embeddings for each layer are produced via global average pooling. Additional details are given in Appendix A.2. We use the linear probes to measure the strength of the tunnel effect.

In [9], authors showed that OOD accuracy decreased in the tunnel, whereas ID accuracy showed a monotonically increasing trend. However, they did not evaluate the strength of the tunnel and defined the start of the tunnel as when the ID probe reached either $95\%$ or $98\%$ of the final ID accuracy. Instead, we propose three metrics that enable us to measure the tunnel's strength, which are tied to OOD accuracy rather than ID accuracy. These metrics are computed for each OOD dataset. Our findings suggest that defining the start of the tunnel based on ID accuracy is ineffective. In Fig. 1, ID accuracy for $32 \times 32$ images reaches $95\%$ of the final accuracy at layer 14, while OOD accuracy degrades starting at layer 9. We use normalized accuracy curves in the main text to compare models across OOD datasets and resolutions, where each linear probe curve is divided by the highest value.

**Metric 1: % OOD Performance Retained.** We measure the magnitude of the tunnel effect by evaluating the OOD performance through layer-wise linear probing. For a given OOD dataset, and a network with $N$ layers, we find the layer that achieves the highest OOD linear probe accuracy, denoted as $l_m$. The linear probe accuracy of $l_m$ is denoted as $a_m$. Then, we denote the OOD accuracy of the linear probe at the penultimate layer $l_{N-1}$ as $a_p$. We assume that $l_m$ is the start of the tunnel if $l_m < l_{N-1}$ and $a_m > a_p$. For an OOD dataset, the % OOD performance retained, $r$, w.r.t. the start of the tunnel is defined as $r = 100 \times (a_p/a_m)$. Higher $r$ means better OOD generalization, hence a weaker tunnel and vice-versa. When $a_p = a_m$, there is no tunnel.

**Metric 2: Pearson Correlation.** Linear probe accuracy of both ID and OOD datasets should display similar trends (higher correlation) in the extractor layers. However, they should have a low correlation for the tunnel. To quantify this, we use Pearson correlation ($\rho$) between ID and OOD accuracy curves, where a higher correlation indicates less tunnel effect. Additional details are given in Appendix A.8.

**Metric 3: ID/OOD Alignment.** A strong model should have higher ID and OOD accuracy, whereas a weak model may show poor accuracy on ID or OOD or both. To capture the characteristic of a model in terms of how strongly it performs on both ID and OOD, we introduce a metric, called ID/OOD Alignment, denoted by $\mathcal{A}$. To formalize, we denote ID and OOD accuracy by $\alpha_{id}$ and $\alpha_{ood}$, respectively. Chance accuracy (random guess) for ID and OOD datasets are denoted by $c_{id}$ and $c_{ood}$, respectively. Finally, we define the metric as $\mathcal{A} = (\alpha_{id} - c_{id}) \times (\alpha_{ood} - c_{ood})$, where $\mathcal{A}, \alpha_{id}, \alpha_{ood}, c_{id}, c_{ood} \in [0, 1]$. Higher $\mathcal{A}$ means greater alignment between ID and OOD accuracy.

### 3.2 Variables Investigated for Their Role in OOD Generalization

We study the role of image augmentation, training classes, training samples, image resolution, and DNN architecture on the tunnel effect, with the details for each given in the next paragraphs.

**Augmentation.** Prior work [22, 23] studied the impact of augmentation independently on OOD generalization whereas its impact on the tunnel effect has not been studied. To address this, we conducted *512 experiments*, where half of them were trained with augmentations and half without. These are done for every combination of the other variables we study. We used random resized crop and random horizontal flip augmentations.

**Number of Classes.** In [9], the tunnel effect was shown to decrease as the number of classes and training samples increased. We aim to disentangle these two variables. We conducted *48 experiments* with ImageNet-100, where we kept the training set fixed at 10,000 samples but varied the class counts: 10 (1000 images per class), 50 (200 images per class), and 100 (100 images per class).

**Number of Samples.**  We conducted *64 experiments* using ImageNet-100 to assess the impact of the number of samples on the tunnel effect. We varied the number of training data per class from 100, 200, 500, and 700 while keeping the number of classes fixed at 100.

**Resolution.**  Since [9] only studied $32 \times 32$ image datasets, the impact of image resolution on the tunnel effect is unknown. We hypothesized that higher resolutions would result in more hierarchical features, resulting in reducing the tunnel effect. To test this, we trained models on ImageNet-100 with $32 \times 32$, $64 \times 64$, $128 \times 128$, and $224 \times 224$ images, *while keeping the number of parameters for each architecture constant.* We conduct 48 experiments per resolution (*192 total*).

**DNN Architecture Variables.**  We study the tunnel effect in eight DNN architectures drawn from three families: VGG [53], ResNet [54], and ViT [55]. We study the role of the size of the $k \times k$ *stem*, which is the size of the first CNN filter or the ViT's patch size. Because over-parameterization is central to the tunnel effect hypothesis, we measure the *over-parameterization level* [56], $\gamma = P/N$, where $P$ is the number of DNN parameters and $N$ is the number of ID training samples. We conducted *416 experiments* to assess the impact of architecture type, depth, over-parameterization level, stem style, and spatial reduction.

We ensure that each DNN architecture uses the *same* number of parameters across image resolutions. To do this for the VGG family, we created VGGm, which replaces the two fully connected layers before the output layer with a ResNet-style global average pooling layer. Since the original VGG is designed for high-resolution images ($224 \times 224$), it includes the max-pool in all 5 stages to progressively reduce the spatial dimension, $s$ of the features ($s \times s \times c$) across VGG layers. To capture this, we introduce a variable named *spatial reduction* by which a stem layer (first layer) reduces the spatial dimension of input images. Spatial reduction, $\phi$ is defined as the ratio of the output spatial dimension $s_{out}$ to the input spatial dimension $s_{in}$, i.e., $\phi = s_{out}/s_{in}$. For instance, spatial reduction at a layer that reduces the spatial dimension from $32 \times 32$ into $16 \times 16$ becomes 0.5, whereas a DNN that did no spatial reduction would have $\phi = 1$. We also created another variant, VGGm†, to study the impact of spatial reduction on the tunnel effect. The difference between VGGm ($\phi = 0.5$) and VGGm† ($\phi = 1$) is that VGGm includes max-pool in all 5 stages whereas VGGm† omits max-pool in the first 2 stages for $32 \times 32$ input resolution. Compared to VGGm, VGGm† achieved higher ID accuracy on ImageNet-100 (see Table 11). For ResNet, we use the original ResNet architecture [54]. To keep model size constant across resolutions for ViT models, we use a fixed patch size of $8 \times 8$, with the number of patches being larger for higher-resolution images. Following [57], we used 2D sin-cos position embeddings to encode spatial information.

## 3.3  Datasets

**ID Datasets.**  In our main experiments, we train DNNs on 3 ID datasets: 1) *ImageNet-100* [58]—a subset (100 classes) of ImageNet-1K, 2) *CIFAR-10* [59], and 3) and *CIFAR-100* [60]. For these experiments, 52 DNNs were trained on ImageNet-100, 8 on CIFAR-100, and 4 on CIFAR-10 (*64 DNNs total*), where resolution, augmentation, etc., were varied as described earlier. ID and OOD linear probes are trained and evaluated for each DNN layer. For our experiments on downloaded ImageNet-1K pre-trained DNNs, ID linear probes were trained on a training subset consisting of 50 images per class (50,000 images). Standard test sets are used for all ID datasets.

**OOD Datasets.**  To assess OOD generalization with linear probes, we use 9 OOD datasets: *NINCO* [61], *ImageNet-R* [62], *CIFAR-100* [60], *CIFAR-10* [59], *Oxford 102 Flowers* [63], *CUB-200* [64], *Aircrafts* [65], *Oxford-IIIT Pets* [66], and *STL-10* [67] (see Appendix B). Eight OOD datasets are used with DNNs trained on each ID dataset, where CIFAR-10 is omitted for DNNs trained on ImageNet variants. When using CIFAR-10 or CIFAR-100 as the ID dataset, the other CIFAR dataset is used for OOD experiments since their classes do not overlap.

**Resolution & ID Accuracy.**  All DNNs trained on CIFAR-10 or CIFAR-100 are trained and evaluated with $32 \times 32$ images. For DNNs trained on ImageNet variants, all ID and OOD images were resized to the resolution with which the DNN was trained. See Table 11 for the resolution used for each DNN and their ID accuracy with and without augmentations.

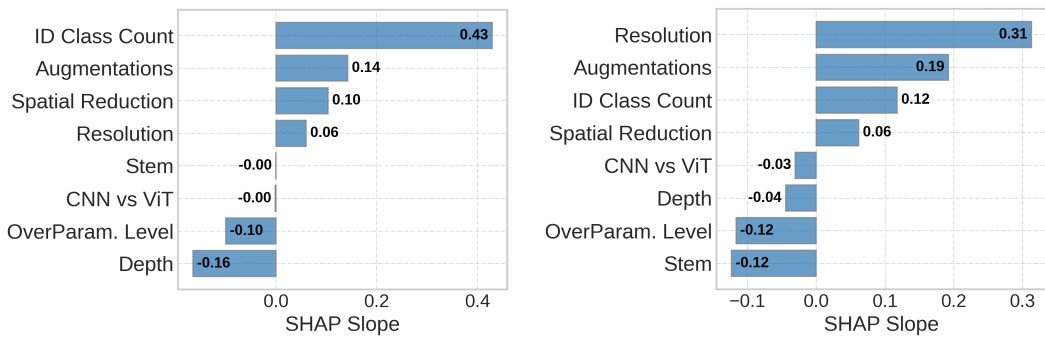

(a) % OOD performance retained as target

(b) ID/OOD alignment as target

Figure 2: **SHAP Results.** SHAP slope shows the individual contribution of variables to various targets. Positive values indicate enhanced OOD generalization, and vice-versa for negative values.

## 3.4 Statistical Analysis

For each of the 64 DNNs trained on an ID dataset in our main results, we compute our OOD generalization metrics on each OOD dataset, resulting in 512 values per metric. These values are derived from 8,604 linear probes (ID and OOD). We study the impact of each variable on OOD generalization in isolation using paired Wilcoxon signed-rank tests at $\alpha = 0.05$, where pairs are constructed to control for the impact of other variables, and we use Cliff's Delta to measure effect sizes [68], which is appropriate for ratio data. For Cliff's Delta ($\delta$), we follow the standard practice of defining a negligible effect for $|\delta| < 0.147$, small effect for $0.147 \leq |\delta| < 0.33$, medium effect for $0.33 \leq |\delta| < 0.474$, and large effect for $|\delta| \geq 0.474$.

To jointly analyze and rank the contribution of each variable, we use SHAP [†], which determines the contribution of each input variable to its output [69, 70]. Following [71], we train Gradient Boosting Regression models to predict *three output targets*: a) % OOD performance retained, b) Pearson correlation, and c) ID/OOD alignment, from *8 input variables*: 1) resolution, 2) augmentation, 3) ID class count, 4) spatial reduction, 5) stem, 6) CNN vs. ViT, 7) over-parametrization level, and 8) depth. We then obtain SHAP values for each variable, where using Gradient Boosting Regression facilitates controlling for variable interaction effects [71]. Because SHAP magnitude does not indicate the direction of a variable's impact, for each of the 3 models, we fit a linear regression model between each variable and its corresponding SHAP values to obtain its slope. Positive slopes indicate the variable improves the metric. We call this *SHAP Slope.* Details are given in Appendix A.9.

## 4 Experiments & Results

### 4.1 Main Experiments

In [9], all architectures were trained with CIFAR-10 or CIFAR-100, and all experiments were conducted with $32 \times 32$ images. Instead, most of our experiments use ImageNet-100 as the ID dataset, where image resolution is varied. We also include experiments on CIFAR datasets. In our main experiments, for each of our 64 DNNs, we produced 8 OOD linear probe curves and 1 ID linear probe curve (576 total), which required computing *8604 linear probes*. From this, we obtain 512 values for each of our 3 metrics under various conditions.

#### 4.1.1 Overall Findings & SHAP Analysis

Results for our SHAP Slope analysis computed across all 512 OOD experiments are summarized in Fig. 2, which shows the impact of each variable on the % OOD performance retained and ID/OOD alignment. The SHAP Slope figure for Pearson Correlation is given in Appendix C since it has nearly identical trends to % OOD performance retained. The $R^2$ for % OOD performance retained, ID/OOD alignment predictions, and Pearson Correlation are 0.62, 0.44, and 0.73, respectively. Each variable's

---

[†] `https://github.com/shap/shap`

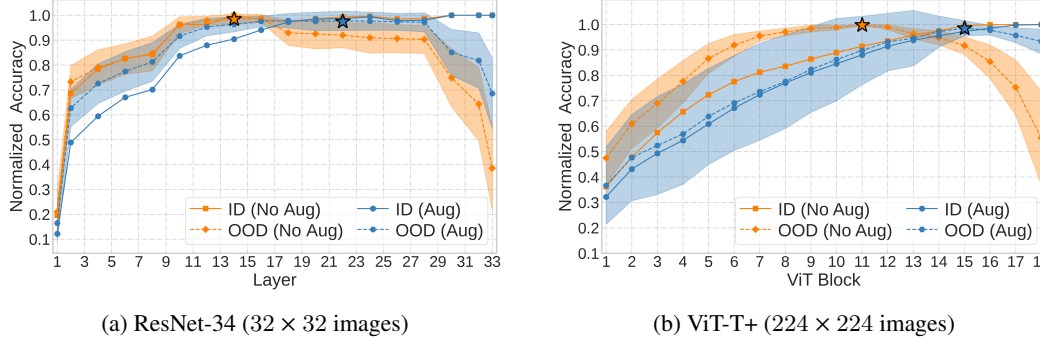

(a) ResNet-34 ($32 \times 32$ images)          (b) ViT-T+ ($224 \times 224$ images)

Figure 3: **Augmentation greatly reduces the tunnel effect.** In (a), augmentation shifts the tunnel from layer 14 to 22, and in (b) from block 11 to 15. The OOD curve is the average of 8 OOD datasets with a shaded area indicating a 95% confidence interval. ☆ denotes the start of the tunnel.

positive/negative impact is consistent across all 3 of our SHAP analyses. Our main findings are given below, with additional findings in Appendix C.

Our metrics reveal that increasing the ID class count, higher resolutions, and using augmentations improve OOD generalization. For % OOD performance retained and Pearson Correlation, increasing the number of ID classes had the greatest impact, whereas, for ID/OOD alignment, increasing resolution did. This is likely because ID/OOD alignment curves are not normalized using the best OOD accuracy, resulting in resolution's role being obscured for the other two metrics. Across metrics, using augmentations had the second greatest positive impact. These results indicate that increasing between-class diversity (more classes), greater within-class diversity (augmentations), and higher image resolutions improve OOD generalization.

While [9] argued that the primary source of the tunnel effect is over-parameterization, our results indicate it plays a minor role compared to other factors. Our results indicate that using a larger stem and excessive DNN depth somewhat impair generalization. For all metrics, the choice of ViT or CNN had the least impact on OOD generalization, consistent with the hypothesis that much of the reported benefits of ViTs for image classification are due to training with larger datasets, stronger augmentation policies, and other tricks [72].

Using the average % OOD performance retained across the 8 OOD datasets to analyze all 64 of our DNNs, 4 had negligible (non-existent) tunnels, 8 had weak tunnels, 13 had medium tunnels, and 39 had strong tunnels. We use intervals of [100%, 95%] for negligible, [90%, 95%) for weak, [80%, 90%) for medium, and [0%, 80%) for strong tunnel. This demonstrates that the tunnel effect is not universal and depends on variables. Next, we dive into the factors that influence tunnel effect strength.

### 4.1.2 Augmentation Results

In Fig. 3, example linear probe plots illustrate that augmentations play a major role in reducing the tunnel effect. To further analyze the impact of augmentation on OOD generalization, we compared all of our experiments in which augmentation was used or omitted with all other variables controlled using paired Wilcoxon Signed-Rank tests (256 paired experiments, 512 total). For % OOD performance retained, augmentations significantly decreased the tunnel effect with 64.26% retained without augmentations and 78.41% with ($p < 0.001$), which had a *medium* effect size ($|\delta| = 0.370$). For Pearson correlation, augmentations also had a significant effect where $\rho$ increased from 0.77 to 0.86 ($p < 0.001$), with a *medium* effect size ($|\delta| = 0.374$). For ID/OOD alignment, augmentations increased alignment from 0.15 to 0.25 ($p < 0.001$), with a *medium* effect size ($|\delta| = 0.357$).

### 4.1.3 Resolution Results

Illustrative examples showing how increasing resolution improves OOD generalization are given in Fig. 1. To study resolution further, we conducted paired tests between models trained with $32 \times 32$ images and those trained with $64 \times 64$, $128 \times 128$, or $224 \times 224$ images (48 paired experiments per resolution comparison, 192 total). All models were trained on ImageNet-100. Mean effect sizes ($\delta$) and $p$-values are given in Table 1. Average values for the 3 metrics computed for each resolution are

Table 1: **Higher resolution images reduce the tunnel effect.** Pairwise statistical analysis between DNNs trained on ($32 \times 32$) images vs. higher resolution images.

| Resolution | % OOD Perf. Retained | | Pearson Correlation | | ID/OOD Alignment | |
|---|---|---|---|---|---|---|
| | Eff. Size ($|\delta|$) ↑ | $p-$val↓ | Eff. Size ($|\delta|$) ↑ | $p-$val ↓ | Eff. Size ($|\delta|$) ↑ | $p-$val ↓ |
| $32^2$ vs $64^2$ | $negli.$ (0.002) | 0.315 | $negli.$ (0.023) | 0.572 | $small$ (0.326) | < 0.001 |
| $32^2$ vs $128^2$ | $small$ (0.171) | 0.001 | $small$ (0.240) | 0.006 | $large$ (0.567) | < 0.001 |
| $32^2$ vs $224^2$ | $small$ (0.198) | 0.005 | $small$ (0.280) | 0.011 | $large$ (0.625) | < 0.001 |

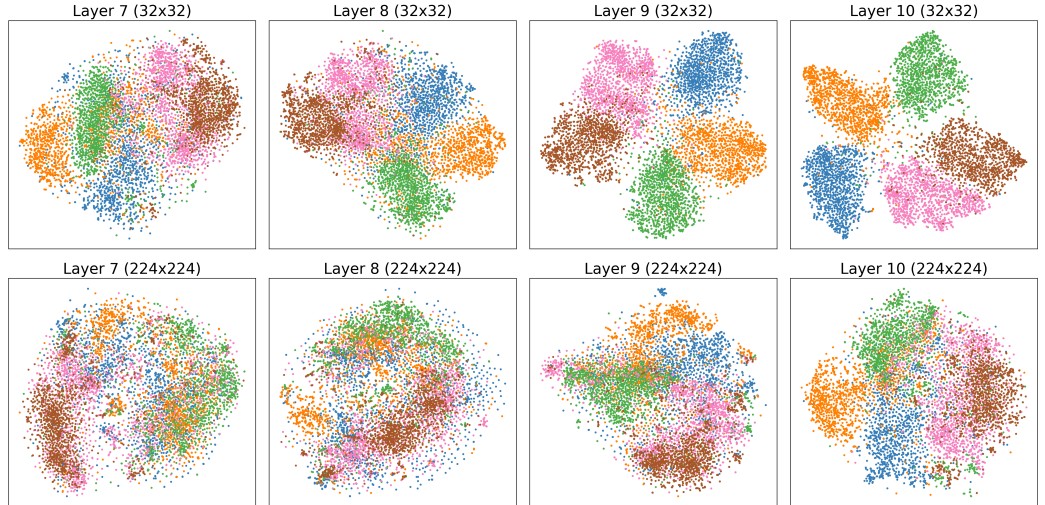

Figure 4: **High-resolution model does not exhibit representation compression.** The t-SNE comparison between VGGm-11 models trained on low- (1st row) and high-resolution (2nd row) images of the same ID dataset (ImageNet-100) in an augmentation-free setting. Layer 8 marks the start of the tunnel in VGGm-11 trained on $32 \times 32$ images whereas $224 \times 224$ resolution does not create any tunnel. Layer 10 is the penultimate layer. The tunnel layers (layers 8-10) progressively compress representations for $32 \times 32$ resolution whereas corresponding layers for $224 \times 224$ resolution do not exhibit similar compression. For clarity, we show 5 classes from ImageNet-100 and indicate each class by a distinct color. The formation of distinct clusters in the $32 \times 32$ model is indicative of representation compression and intermediate neural collapse [10], which impairs OOD generalization.

given in Table 12. The findings are consistent with our SHAP analysis: training on higher-resolution images improves OOD generalization, whereas low-resolution datasets increase tunnel effect strength. Additional results are given in the Appendix C.13.

A t-SNE analysis for various layers from VGGm-11 models trained on $32 \times 32$ and $224 \times 224$ images is given in Fig. 4. The low-resolution model exhibits much greater intermediate neural collapse [10] and representation compression than the high-resolution model. This is likely why many OOD detection algorithms that work well for CIFAR fail for higher-resolution datasets [11]. These results highlight the dangers of extrapolating findings from low-resolution datasets to all of deep learning.

### 4.1.4 DNN Architecture Results

**Spatial Reduction.** Our SHAP analysis revealed that lower values for spatial reduction ($\phi$) hurt OOD generalization. To further study this, we conducted paired tests between VGGm-11 and VGGm-17, which both have $\phi = 0.5$, and VGGm†-11 and VGGm†-17 ($\phi = 1.0$). All 4 DNNs are trained on ImageNet-100 at $32 \times 32$ resolution, with and without augmentations, and each is evaluated on the 8 OOD sets (32 paired experiments, 64 total). In terms of % OOD performance retained, the VGGm† models retained 84.40% whereas the VGGm models retained 64.85% ($p < 0.001$), with a *large* effect size ($|\delta| = 0.531$). Similarly, Pearson correlation significantly decreased from 0.92 to 0.72 ($p < 0.001$), with a *large* effect size ($|\delta| = 0.536$), and ID/OOD alignment significantly decreased from 0.26 to 0.18 ($p < 0.001$), with a *medium* effect size ($|\delta| = 0.361$). Fig. 5 provides example

normalized accuracy curves. VGGm-11 exhibits a strong tunnel spanning from layer 7 to 10 (Fig. 5a), whereas no tunnel is present in VGGm†-11 (Fig. 5b).

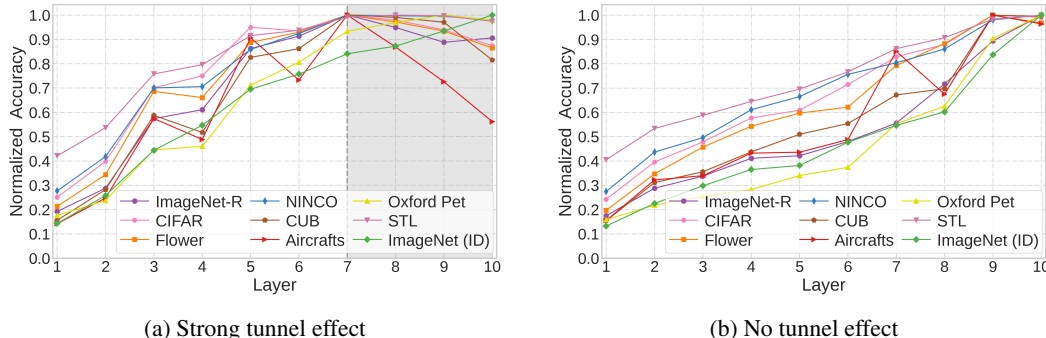

(a) Strong tunnel effect             (b) No tunnel effect

Figure 5: **The tunnel effect is not universal.** In **(a)**, VGGm-11 consisting of max-pool in all 5 stages ($\phi = 0.5$), creates a tunnel (layers 7-10, gray-shaded area). In **(b)**, the same VGGm-11 without max-pool in the first 2 stages ($\phi = 1$, called VGGm†-11), eliminates the tunnel for all OOD datasets.

**Stem.** Our SHAP results indicate that increasing stem size hurts OOD generalization. To further study this, we conducted a paired test over 64 paired experiments between ResNet-18, which uses a $7 \times 7$ stem, and VGGm-17, which uses a $3 \times 3$ stem. Increasing the stem from 3 to 7 significantly decreased the % OOD performance retained from 76.74% to 66.66% ($p < 0.001$), with a *small* effect size ($|\delta| = 0.306$). However, for Pearson correlation, there was no significant difference ($p = 0.145$). For ID/OOD alignment, increasing the stem significantly reduced the score from 0.27 to 0.21 ($p < 0.001$), with a *small* effect size ($|\delta| = 0.226$). Fig. 12 provides box plots of % OOD performance retained and ID/OOD alignment for the three stem values.

**Depth.** Our SHAP analysis revealed that increasing depth impairs OOD generalization. As shown in Fig. 10, increasing depth impairs OOD performance retention for each architecture family. To study this further, we compared VGGm-11 and VGGm-17 using 48 paired experiments (96 total). Increasing depth significantly decreased % OOD performance retained from 89.19% to 69.41% ($p < 0.001$), with a *large* effect size ($|\delta| = 0.539$). Likewise, Pearson correlation significantly decreased from 0.94 to 0.80 ($p < 0.001$), with a *large* effect size ($|\delta| = 0.497$). ID/OOD alignment also significantly decreased from 0.28 to 0.25 ($p < 0.001$) with a *small* effect size ($|\delta| = 0.161$).

**Over-parameterization Level.** Our SHAP analysis showed that over-parameterization level negatively impacts OOD generalization. Fig. 11 shows how increasing the over-parameterization level decreases % OOD performance retained. We conducted paired tests between VGGm-11 ($\gamma = 74.7$) and ResNet-34 ($\gamma = 168.4$) to study this further (32 paired experiments). Increasing over-parameterization significantly reduced % OOD performance retained from 87.22% to 62.78% ($p < 0.001$), with a *large* effect size ($|\delta| = 0.680$). Likewise, the Pearson correlation was significantly reduced from 0.93 to 0.82 ($p < 0.001$), with a *large* effect size ($|\delta| = 0.570$). Lastly, ID/OOD alignment significantly dropped from 0.29 to 0.20 ($p < 0.001$), with a *medium* effect size ($|\delta| = 0.340$).

### 4.1.5 ID Dataset Size vs. Total Classes

Our SHAP analysis shows that ID class count positively impacts OOD generalization. To further analyze this, we trained VGGm-11 on different subsets of ImageNet-100 with $32 \times 32$ images where we kept the training dataset fixed at 10,000 samples but varied the class counts: 10 (1000 samples per class), 50 (200 samples per class), and 100 (100 samples per class). Experiments were done with and without augmentations. As shown in Fig. 6, increasing the number of classes greatly reduces the tunnel effect (Figs. 6a and 6b). To examine the role of the dataset size, we vary the number of samples per class from 100, 200, 500, and 700 while keeping the class count constant at 100, which has a relatively small impact on the tunnel effect (Figs. 6c and 6d).

### 4.2 Analysis of Widely Used Pre-trained Backbones

We also studied OOD generalization for eight widely used ImageNet-1K pre-trained CNN and ViT backbones trained with either supervised learning (SL) or self-supervised learning (SSL). We studied

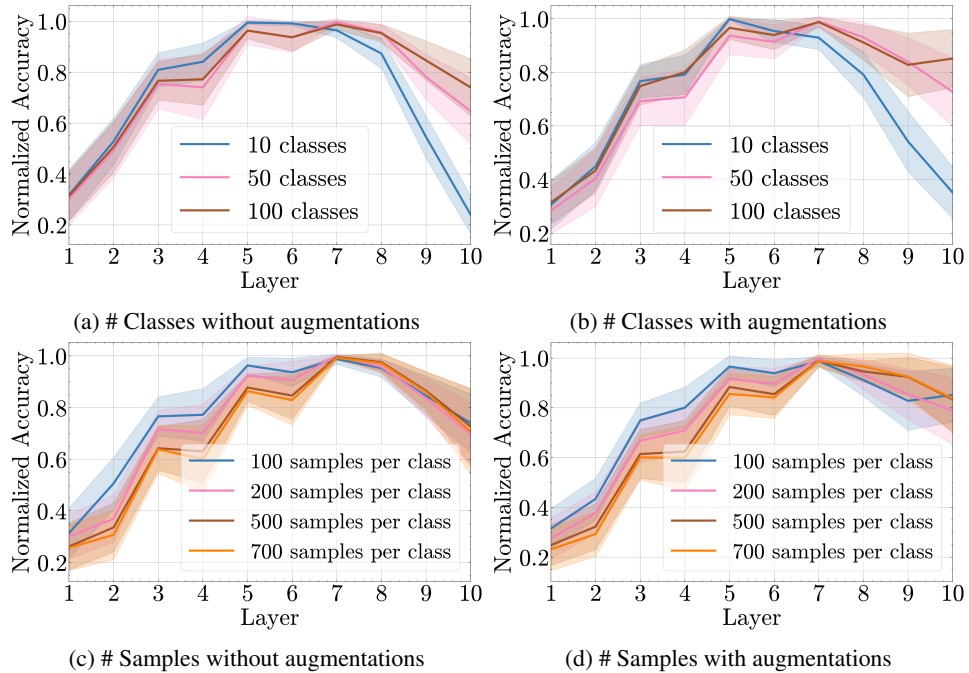

(a) # Classes without augmentations

(b) # Classes with augmentations

(c) # Samples without augmentations

(d) # Samples with augmentations

Figure 6: **Training on more classes greatly reduces the tunnel effect, whereas increasing dataset size has less impact. (a)** and **(b)** Results with a fixed number of samples but a varied number of classes. **(c)** and **(d)** Results with a fixed number of classes but a varied number of samples per class.

ResNet-50 (1 SL and 1 SSL [73]), 4 ViT-B (1 SL and 3 SSL [50, 74, 75]), and ConvNeXt-B (1 SL and 1 SSL [76]) models. We trained linear probes on ImageNet-1K (ID) and our 8 OOD datasets (*1980 linear probes total*), resulting in 72 values per OOD metric. As shown in Fig. 17 and Table 9, the tunnel effect is absent in most models and is weakly present in both SL and SSL ResNet-50. Additional results are given in Appendix C.11. Appendix A.6 includes implementation details.

### 4.3 Continual Learning Results

Catastrophic forgetting is a central focus in continual learning [17, 77]. Many methods assume forgetting occurs mostly in later layers with earlier layers serving as universal features [16, 78, 79]. The original tunnel effect paper challenged this [9]. They trained VGG-19 on two tasks sequentially, where the first task included half of the CIFAR-10 classes and the second task included the other half. After training on each task, the tunnel and extractor were saved. They found no forgetting when tunnels were swapped, indicating no learning occurred in the tunnel for task 2, and they found reduced forgetting when fine-tuning the extractor alone. These findings are worth assessing beyond CIFAR-10. We ask: *What is the role of the tunnel in mitigating forgetting?*

We replicated their general approach by training ResNet-18 on ImageNet-100, where the first task had the first 50 classes and the second had the rest. After learning each task $t$, we saved the extractor $E_t$, tunnel $T_t$, and the classification head. We conducted this experiment with $32 \times 32$ and $224 \times 224$ images. Using the tunnel definition from [9], tunnels $T_1$ and $T_2$ correspond to layers 14-17 for $32 \times 32$ images and 15-17 for $224 \times 224$ images. See Appendix A.7 for additional details.

Results are given in Table 2. Unlike the findings in [9], when we swapped tunnels $T_1$ and $T_2$, accuracy was greatly reduced for both resolutions, indicating that essential learning happened within the tunnel. Next, we replicated their fine-tuning experiments. If fine-tuning $E_2$ alone reduces forgetting more than fine-tuning $E_2$ with $T_1$, it suggests the tunnel has a detrimental impact on mitigating forgetting. We found that fine-tuning $E_2$ alone improved task 1 accuracy less than fine-tuning $E_2 + T_1$, indicating $T_1$ helps reduce forgetting. Our findings, which contradict [9], suggest the "tunnel" plays an essential role in mitigating catastrophic forgetting, consistent with prior work [5].

Table 2: **Continual Learning Results**. The tunnel is task-specific and impacts forgetting. $E_t$ and $T_t$ denote the extractor and tunnel, respectively, for tasks $t \in \{1, 2\}$. Orange indicates original task accuracy. Red and blue indicate degraded or maximally enhanced accuracy, respectively. Fine-tuning (FT) aims to recover 1st task performance (gray); therefore, the 2nd task columns are empty.

| Combination | $32 \times 32$ Resolution | | $224 \times 224$ Resolution | |
| --- | --- | --- | --- | --- |
| | 1st Task ↑ | 2nd Task ↑ | 1st Task ↑ | 2nd Task ↑ |
| $E_1 + T_1$ | 64.04 | 21.20 | 74.36 | 23.24 |
| $E_1 + T_2$ | 37.64 (↓ 26.40) | 35.64 | 54.32 (↓ 20.04) | 35.68 |
| $E_2 + T_2$ | 20.44 | 68.16 | 20.44 | 78.00 |
| $E_2 + T_1$ | 29.84 | 23.84 (↓ 44.32) | 21.56 | 46.08 (↓ 31.92) |
| $E_2 + T_1(FT)$ | 51.40 (↑ 30.96) | – | 57.24 (↑ 36.80) | – |
| $E_2(FT)$ | 45.60 (↑ 25.16) | – | 52.48 (↑ 32.04) | – |

## 5 Discussion & Conclusions

We conducted extensive experiments to investigate the generality of the tunnel effect in a wide range of transfer settings. Our study indicates that the best way to mitigate the tunnel effect, and thereby increase OOD generalization, is to increase diversity in the ID training dataset, especially by increasing the number of semantic classes, using augmentations, and higher-resolution images; hence, we revise the tunnel effect hypothesis as follows:

> **The Tunnel Effect Hypothesis:** An overparameterized $N$-layer DNN forms two distinct groups:
>
> 1. *The extractor* consists of the first $K$ layers, creating linearly separable representations.
> 2. *The tunnel* comprises the remaining $N - K$ layers, compressing representations and hindering OOD generalization.
>
> $K$ is proportional to the diversity of training inputs, where if diversity is sufficiently high, $N = K$.

Earlier works on the tunnel effect [9] and intermediate neural collapse [10] exclusively used $32 \times 32$ images for ID training. We found that while DNNs trained on CIFAR always exhibited the tunnel effect, the tunnel effect was greatly reduced for ImageNet-100 at higher resolutions. This discrepancy helps explain why methods validated on CIFAR and similar datasets may not generalize in many scenarios [11–18]. We urge the community to use high-resolution datasets with 100 or more categories to improve the generality of findings, especially for studies related to representation learning, neural collapse, and OOD detection/generalization.

**Limitations & Future Work.** While our work validates the existence of tunnels, future research should focus on developing theoretical frameworks to help explain the tunnel effect. We rigorously studied OOD generalization on vision datasets with supervised learning. Future work could study non-vision, multi-modal [80], and biased datasets [81, 82], where the tunnel effect has not yet been studied. While we studied pre-trained SSL backbones, we could not do our SHAP analysis without having more SSL backbones. Valuable insights for SSL could be obtained by conducting carefully controlled paired experiments, as done in our main experiments. This would require probing at least four different SSL algorithms for each variable analyzed, where training each SSL DNN would require over 10× more time than our supervised DNNs. Additionally, SSL methods employ more advanced augmentation policies than ours, where we used random-resized crops and horizontal flips. Replicating our SHAP analysis with multiple augmentation policies could reveal whether the OOD generalization capabilities of SSL algorithms are due to their augmentation policies versus their objective functions. Lastly, identifying regularizers or other techniques that mitigate tunnel formation should be sought for continual learning methods that start from scratch using small initial sets. This could greatly improve forward transfer, leading to more efficient continual learning methods [83–86].

## Acknowledgments and Disclosure of Funding

This work was partly supported by NSF awards #2326491, #2125362, and #2317706. The views and conclusions contained herein are those of the authors and should not be interpreted as representing any sponsor's official policies or endorsements. We thank Junyu Chen for feedback on an early draft of this manuscript.

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

# Appendix

We organize additional implementation details and supporting results as follows:

- Appendix A describes the implementation details and hardware used for training. It describes the DNN architectures (VGG, ResNet, and ViT), feature extraction for linear probing, training, and evaluation details of both pre-training and linear probing in various experiments. The implementation details of continual learning experiments and analysis of the widely used pre-trained models are also described. The implementation details of the SHAP slope analysis are also provided.

- Appendix B provides details on the datasets used in this paper. We used a total of 4 ID datasets and 9 OOD datasets.

- Appendix C presents additional supporting results for the Pearson correlation metric. It includes additional SHAP results. It reports the performance of various models in terms of OOD accuracy, % OOD performance retained, Pearson correlation, and ID/OOD alignment in various settings. It presents analyses of how variables impact the tunnel effect and OOD generalization.

- Appendix D reports the in-distribution performance of various models (VGGm, VGGm†, ResNet, ConvNeXt, and ViT) on various ID datasets (ImageNet-100, ImageNet-1K, CIFAR-10, CIFAR-100).

- Appendix E reports additional statistical results for different variables and interventions.

- Appendix F includes the list of 100 classes in the imageNet-100 dataset and confirms there is no overlap between ID and OOD datasets.

## A  Implementation Details & Computational Resources

In this section, we use several acronyms such as **WD** : Weight Decay, **LR** : Learning Rate, **GAP** : Global Average Pooling, **SSL** : Self-Supervised Learning, **SL** : Supervised Learning, **ID** : In-Distribution, and **OOD** : Out-Of-Distribution. We implemented our code in Python using PyTorch.

### A.1  DNN Architectures

To assess how the depth of DNN affects the tunnel effect, we create a deeper variant of each DNN architecture by increasing the number of layers (CNN) or blocks (ViT) while keeping other configurations identical.

**VGGm.** We modify the VGG-13 (or VGG-19) architecture to create our VGGm-11 (or VGGm-17). This is done by adding an adaptive average pooling layer (*nn.AdaptiveAvgPool2d*), which allows the network to accept any input size while keeping the output dimensions the same. Additionally, we include a single fully connected (FC) layer rather than three fully connected layers, resulting in the VGGm-11 (or VGGm-17) structure. In particular, VGGm has the final FC classifier layer without the additional two FC layers before the final layer. The number of parameters of VGGm-11 (or VGGm-17) is the same across all input resolutions.

**ResNet.** To maintain the same number of parameters across all input resolutions, we use the original and unmodified ResNet-18/34 [54]. Thus ResNet-18/34 consists of $7 \times 7$ convolution with stride 2 in the stem layer (first layer) followed by a maxpool layer.

**ViT.** We select ViT-Tiny (5.61M parameters) for our main experiments with varying resolutions. To maintain exact same number of parameters across all resolutions, we use a fixed patch size of $8 \times 8$ with a varied number of patches due to spatial dimensions. The number of patches or image tokens for $32 \times 32$ and $224 \times 224$ resolutions are 16 and 784 respectively. We use an embedding dimension of 192, a depth of 12 (i.e., 12 ViT blocks) and 3 heads. We also created another deeper variant with a depth of 18 (i.e., 18 ViT blocks), referred to as ViT-Tiny+ (8.39M parameters). To maintain the same number of parameters across all input resolutions, following [57], we omit the learnable position embeddings and instead use the fixed 2D sin-cos position embeddings. Other details adhere to the original ViT paper [55].

### A.2 Feature Extraction For Linear Probing

In experiments with CNNs, at each layer $l$, for each sample, we extract features of dimension $H_l \times W_l \times C_l$, where $H_l$, $W_l$, and $C_l$ denote the height, width and channel dimensions respectively. Next, following [37], we apply $2 \times 2$ adaptive average pooling on each spatial tensor ($H_l \times W_l$). After average pooling, features of dimension $2 \times 2 \times C_l$ are flattened and converted into a vector of dimension $4C_l$. Finally, a linear probe is trained on the flattened vectors. In experiments with ViTs, following [87], we apply global average-pooling (GAP) to aggregate image tokens excluding the class token and train a linear probe on top of GAP tokens.

### A.3 VGGm Experiments

**VGGm ID Training:** For training VGGm-11/17 on ImageNet-100, we employ the AdamW optimizer with an LR of $6 \times 10^{-3}$ and a WD of $5 \times 10^{-2}$ for batch size 512. The model is trained for 100 epochs using the Cosine Annealing LR scheduler with a linear warmup of 5 epochs. We use label smoothing of 0.1 with cross-entropy loss. For CIFAR-10 and CIFAR-100 datasets, we use an LR of 0.01 for batch size 512. We train VGGm-11 for 100 and 70 epochs in experiments without augmentations and with augmentations respectively. Whereas VGGm-17 is trained for 100 epochs in both settings without augmentations and with augmentations.

**Varying Number of Classes and Sample Size:** VGGm-11 uses LR of $8 \times 10^{-3}$ and WD of $5 \times 10^{-2}$. VGGm-17 uses WD of $1 \times 10^{-2}$ instead of $5 \times 10^{-2}$. For both with and without augmentation, We train VGGm-11 for 100 epochs and VGGm-17 for 200 epochs.

**VGGm Linear Probing:** We use the AdamW optimizer with a flat LR of $1 \times 10^{-3}$ and a WD of $0$ for batch size 128. The linear probes are trained for 30 epochs. We use label smoothing of 0.1 with the cross-entropy loss.

### A.4 ResNet Experiments

**ResNet ID Training:** For training ResNet-18/34, we employ the AdamW optimizer with an LR of 0.01 and a WD of 0.05 for batch size 512. The model is trained using the Cosine Annealing LR scheduler with a linear warmup of 5 epochs. We use label smoothing of 0.1 with cross-entropy loss. We use 100 epochs to train ResNet-18 in experiments without augmentation. In experiments with augmentations, we train ResNet-18 for 80 epochs using random resized crop and random horizontal flip augmentations. The ResNet-34 models are trained for 100 epochs in experiments with and without augmentations.

**ResNet Linear Probing:** In the linear probing experiment, we use the AdamW optimizer with an LR of $1 \times 10^{-3}$ and a WD of $0$ for batch size 128. The linear probes are trained for 30 epochs. We use label smoothing of 0.1 with cross-entropy loss.

### A.5 ViT Experiments

For ViT training, we follow the codebase of DeiT [88] and A-ViT [89]. We also use `timm` library [90].

**ViT ID Training:** For training ViT-Tiny/ViT-Tiny+, we employ the AdamW optimizer with an LR of $8 \times 10^{-4}$ and a WD of 0.05 for batch size 96. We use the Cosine Annealing LR scheduler with a linear warm-up (5 epochs). We also use label smoothing of 0.1 with cross-entropy loss. We train the ViT-Tiny for 100 and 40 epochs in experiments without augmentations and with augmentations respectively. Whereas the ViT-Tiny+ is trained for 100 and 60 epochs in experiments without augmentations and with augmentations respectively. In the experiments with augmentations, we use random resized crop and random horizontal flip. Following [57, 87], we omit class token and instead use GAP token by global average-pooling image tokens for classification head.

**ViT Linear Probing:** We use the AdamW optimizer with LR of 0.01 and WD of $1 \times 10^{-4}$ for batch size 512. The linear probes are trained for 30 epochs. We use label smoothing of 0.1 with cross-entropy loss.

Table 3: Augmentations used for training various large pre-trained models.

| Model | Augmentations |
|---|---|
| SwAV ResNet-50 | Random crop, horizontal flip, color distortion, Gaussian blur |
| MAE ViT-B | Crop, horizontal flip, masking |
| MUGS ViT-B | Random crop, color jitter, Gaussian noise, horizontal flip, gray scaling, auto-aug |
| DINO V1 ViT-B | Color jittering, Gaussian blur, solarization, multi-crop |
| FCMAE ConvNeXt-B | Random crop, RandAug, masking |
| SL ResNet-50 | Random erase, mixup, cutmix, auto-aug |
| SL ConvNeXt-B | Mixup, cutmix, RandAug, Random Erasing |
| SL ViT-B | Mixup, cutmix, auto-aug |

Note that, in all cases, we maintain the over-parameterization level where the DNN model size (number of parameters) exceeds the number of training samples ($\gamma > 1$). After ID training, we select the model or checkpoint that achieves the best top-1 accuracy (%) on the validation dataset.

## A.6 Pre-trained Model Experiments

We describe the implementation details of large pre-trained model analyses that are presented in Sec. 4.2. We list the augmentations used by various pre-trained models in Table 3.

**Datasets.** Since it is computationally expensive to use the full ImageNet-1K (ID) dataset for layer-wise linear probing experiments, we use a subset of ImageNet-1K as the training dataset where we randomly sampled 50 images per class from the ImageNet-1K dataset. During the evaluation, we used the original ImageNet-1K validation dataset (50 images per class). For OOD linear probe experiments, we used the same OOD datasets e.g., CIFAR-100, Flower-102, NINCO, Aircrafts-100, Oxford Pets-37, STL-10, CUB-200, and ImageNet-R.

**ViT.** For all ViT pre-trained models, we use ViT-Base (85.8M parameters) with depth of 12 and 16×16 patch size.

**Pre-trained Model Download Links.** The links to download pre-trained models are given below: *DINO V1 ViT-B* : `https://huggingface.co/timm/vit_base_patch16_224.dino`, *MAE ViT-B*: `https://huggingface.co/timm/vit_base_patch16_224.mae`, *MUGS ViT-B*: `https://huggingface.co/zhoupans/Mugs`, *SwAV ResNet-50* (800 epochs): `https://github.com/facebookresearch/swav`, *FCMAE ConvNeXt-B*: `https://huggingface.co/timm/convnextv2_base.fcmae_ft_in1k`, *ConvNeXt-B* : `https://huggingface.co/facebook/convnext-base-224`, *ResNet-50* (IMAGENET1K_V2): `https://pytorch.org/vision/stable/models.html`, and *ViT-B* (IMAGENET1K_V1): `https://pytorch.org/vision/main/models/generated/torchvision.models.vit_b_16.html`.

**Pre-trained Model Linear Probing.** We attach a linear probe layer to each block of ViT models and each Conv2d layer of CNN models. In the ConvNeXt block, which has three layers, only the 7x7 layer uses Conv2d. Therefore, using the Conv2d layer results in one layer per ConvNeXt block for linear probing. We use the AdamW optimizer with a WD of 0 and a batch size of 512. Linear probes are trained for 30 epochs using cross-entropy loss with label smoothing of 0.1. For all ViT models and ResNet-50 Supervised model, the LR is $5 \times 10^{-4}$. And, for the ResNet-50 SwAV model and both ConvNeXt models, the LR is $3 \times 10^{-5}$.

## A.7 Continual Learning Experiments

We describe the implementation details of continual learning experiments that are presented in Sec. 4.3. We adapt the experimental design from [9] for this study. We use the same linear probing approach and definition of the tunnel to identify tunnels in ResNet-18 models during continual learning experiments. We train the model on two subsequent tasks and each task contains non-overlapping 50 ImageNet classes.

**Continual Learning.** We employ the AdamW optimizer with LR of 0.01 and WD of 0.05 for batch size 512. The model is trained for 50 epochs using the Cosine Annealing LR scheduler with a linear warmup of 5 epochs (for the first task). We use label smoothing of 0.1 with cross-entropy loss.

**Tunnel Layers.** For $32 \times 32$ resolution, tunnels $T_1$ and $T_2$ correspond to ResNet layer 14-17. For $224 \times 224$ resolution, tunnels $T_1$ and $T_2$ correspond to ResNet layer 15-17.

**Finetune.** In this experiment, we use the AdamW optimizer with LR of 0.01 and WD of 5e-2 for batch size 512. The number of epochs is 50. We use label smoothing of 0.1 with cross-entropy loss. A Step LR scheduler is used. In particular, we reduce LR by a factor of 0.1 at predefined milestones, which are set to occur at the 20th and 40th epochs.

In all experiments, we use image augmentations namely random resized crop, random horizontal flip, mixup [91], and cutmix [92]. For mixup and cutmix augmentations, we use participation probability of 0.6 and 0.4 respectively. We set coefficient $\beta = 1.0$ for cutmix and coefficient $\alpha = 0.1$ for mixup.

We adapt image pre-processing steps from [89] for various image resolutions. We use the same number of network parameters, hyperparameters, and identical settings for all image resolutions. For a fair comparison, we choose hyperparameters to maximize the performance of all compared methods or models.

## A.8 Evaluation Metrics

**Accuracy.** For accuracy in all experiments including ID training and ID or OOD linear probing, we use the best top-1 accuracy (%).

**Pearson Correlation.** We compute the Pearson correlation $\rho$ between ID accuracy and OOD accuracy using the following equation,

$$\rho = \frac{\sum_{i=1}^{n}(x_i - \overline{x})(y_i - \overline{y})}{\sqrt{\sum_{i=1}^{n}(x_i - \overline{x})^2 \sum_{i=1}^{n}(y_i - \overline{y})^2}}. \tag{1}$$

In this formula, $x_i$ represents the ID accuracy at the $i$-th layer, and $y_i$ represents the OOD accuracy at the $i$-th layer. $\overline{x}$ and $\overline{y}$ are the average of ID and OOD accuracy over layers, respectively. The index $i$ spans from the first layer (1) to the penultimate layer ($n$) of models since ID and OOD accuracy are computed at each layer. $\rho \in [-1, 1]$, where $\rho = 1$ indicates a perfect positive linear relationship, and $\rho = -1$ indicates a perfect negative linear relationship.

**Wilcoxon Signed-Rank Test.** To measure the significance of the difference between the two groups, we performed a Wilcoxon signed-rank test between the two groups. We compared % OOD performance retained, Pearson correlation, and ID/OOD alignment metrics between each pair. We used the *wilcoxon* function from the *spicy.stats* python library.

**Cliff's Delta.** The effect size (Cliff's Delta) quantifies the statistical difference between the two sets. A bigger effect size denotes a bigger difference and the order is negligible < small < medium < large. We computed Cliff's Delta for % OOD performance retained, Pearson correlation, and ID/OOD alignment metrics to measure the effect size between the two groups. We used the *cliffs_delta* function from the *cliffs_delta* python library (`https://github.com/neilernst/cliffsDelta`).

**Reasons for Multiple Metrics.** The Pearson correlation metric gauges the correlation between ID and OOD accuracy, indicating tunnel behavior (low correlation), but doesn't quantify the tunnel effect's magnitude. The % OOD performance retained metric assesses the tunnel effect's magnitude and identifies tunnel layers but solely compares OOD performance between the highest and penultimate layers, neglecting ID accuracy. The ID/OOD alignment metric addresses this by considering both ID and OOD accuracy. To ensure robustness, we employ all metrics to comprehensively capture ID and OOD performance and their relation to the tunnel effect.

## A.9 SHAP Slope Analysis Details

We describe the implementation details of SHAP slope analyses that are presented in Sec. 4.1.1. For SHAP analysis, categorical variables (e.g., CNN vs ViT) are treated as one-hot vectors, while non-categorical values are transformed into ordinal numbers. We calculate SHAP values on a Gradient Boosting Regression model, which is trained on our set of 512 experiments to predict a metric from the manipulated input variables already described in the main section of this work. We use the Huber loss to train the Gradient Boosting Regression model.

While SHAP values themselves can give us insights into the impact of each variable, their interpretation from commonly used plots like SHAP mean absolute bar plot (Fig. 7a) or SHAP Beeswarm plot

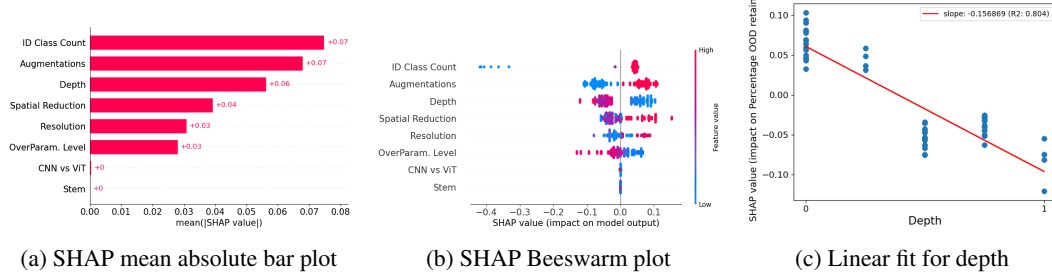

| (a) SHAP mean absolute bar plot | (b) SHAP Beeswarm plot | (c) Linear fit for depth |
|---|---|---|

Figure 7: **Examples of SHAP analysis.** SHAP analysis plots for the % OOD performance retained metric and an example of the slope calculation for the depth. (a) Mean absolute SHAP bar plot, (b) Beeswarm plot, and (c) SHAP values vs normalized ordinal values of depth.

(Fig. 7b) is limited. We want to show both the impact and the relationship direction of each variable with the predicted metric. To do this, for each variable, we fit a linear model on the obtained SHAP values and the normalized ordinal values of the variable. An example can be seen in Fig. 7c. In this way, the value of the slope tells us the impact of the variable on the predicted metric, while the sign of the slope tells us the direction. For each metric, we $L_1$ normalize the slope values. A positive slope means that increasing the variable's value increases the predicted metric, while a negative slope means that increasing the variable's value decreases the value of the predicted metric. A bigger value in any direction indicates that the predicted metric is more sensitive to the corresponding variable. We coined the plot containing the slopes for each variable as the "SHAP Slope" plot.

### A.10 Compute Resources

We trained a total of 64 backbones and 10,652 linear probes to finalize our manuscript. We ran all experiments using four NVIDIA A5000 GPUs, including training backbones and linear probes. The aggregated compute time is ~1,161 (wall clock) hours (48 days).

## B Datasets

All datasets are widely used and publicly available. We provide a link to the license if it exists. Despite their widespread use, we were unable to identify the license for CIFAR-10, CIFAR-100, ImageNet-1K, CUB-200, Aircrafts-100, Flowers-102, and STL-10 datasets.

We used **11 datasets** in total in our paper and they are ImageNet-1K, ImageNet-100, ImageNet-R, CIFAR-10, CIFAR-100, NINCO, CUB-200, Aircrafts, Oxford Pets, Flowers-102, and STL-10. The dataset details are given below:

**CIFAR-10.** CIFAR-10 [59] dataset contains $32 \times 32$ color images with 10 classes. It comprises a total of 60,000 images and each class has 6,000 images. The dataset is split into 50,000 training images and 10,000 test images. CIFAR-10 is a public dataset but does not specify any particular license (`https://www.cs.toronto.edu/~kriz/cifar.html`).

**CIFAR-100.** CIFAR-100 [60] dataset is similar to CIFAR-10 but with 100 classes. And, each class has 600 images. It comprises a total of 60,000 images. The training and test sets contain 50,000 and 10,000 images respectively. Note that the classes in CIFAR-100 are mutually exclusive with those in CIFAR-10. CIFAR-100 is a public dataset but does not specify any particular license (`https://www.cs.toronto.edu/~kriz/cifar.html`).

**ImageNet-1K.** Imagenet-1K [93] is a widely-used large-scale dataset with 1000 object categories and over 1.2 million training images ($224 \times 224$). The test set contains 50000 images. ImageNet-1K dataset is available for free to researchers for non-commercial use. There is no particular license specified by creators (`https://image-net.org/challenges/LSVRC/`).

**ImageNet-100.** ImageNet-100 [58] is a subset of ImageNet-1K and contains 100 ImageNet classes. It consists of 126689 training images ($224 \times 224$) and 5000 test images. The dataset is available to use for research purposes under a BSD 2-Clause License: `https://github.com/HobbitLong/CMC/blob/master/LICENSE`. The class names of ImageNet-100 dataset are given in Sec. F.

**NINCO (No ImageNet Class Objects).** NINCO [61] is a dataset with 64 classes. The dataset is curated to use as an out-of-distribution dataset for ImageNet-1K in-distribution dataset. NINCO dataset has 5878 samples and the classes do not overlap with ImageNet classes. We split 5878 samples into 4702 samples for training and 1176 samples for evaluation. We do not have a fixed number of samples per class for training and evaluation datasets. The dataset is available to use for research purposes under an MIT license: `https://github.com/j-cb/NINCO/blob/main/LICENSE`.

**ImageNet-Rendition (ImageNet-R).** ImageNet-R incorporates distribution shifts using different artistic renditions of object classes from the original ImageNet dataset [62]. We use a variant of ImageNet-R dataset from [94]. ImageNet-R is a challenging benchmark for continual learning, transfer learning, and OOD detection. It consists of classes with different styles and intra-class diversity and thereby poses significant distribution shifts for ImageNet-1K pre-trained models [94]. It contains 200 classes, 24000 training images, and 6000 test images. The dataset is available to use for research purposes under an MIT license: `https://github.com/hendrycks/imagenet-r/blob/master/LICENSE`.

**CUB-200.** CUB-200 is composed of 200 different bird species [64]. The CUB-200 dataset comprises a total of 11,788 images, with 5,994 images allocated for training and 5,794 images for testing. CUB-200 is a public dataset but does not specify any particular license (`http://www.vision.caltech.edu/datasets/cub_200_2011/`).

**Aircrafts-100.** The Aircrafts or FGVCAircrafts dataset [65] consists of 100 different aircraft categories and 10000 high-resolution images with 100 images per category. The training and test sets contain 6667 and 3333 images respectively. Aircrafts dataset is available for non-commercial research purposes only and does not mention a particular license (`http://www.robots.ox.ac.uk/~vgg/data/fgvc-aircraft/`).

**Oxford Pets-37.** The Oxford Pets dataset includes a total of 37 various pet categories, with an approximately equal number of images for dogs and cats, totaling around 200 images for each category [66]. The dataset is available to use for commercial/research purposes under a Creative Commons Attribution-ShareAlike 4.0 International License: `https://www.robots.ox.ac.uk/~vgg/data/pets/`.

**Flowers-102.** The Flowers-102 dataset contains 102 flower categories that can be easily found in the UK. Each category of the dataset contains 40 to 258 images [63]. Flowers-102 is a public dataset but does not mention a particular license (`https://www.robots.ox.ac.uk/~vgg/data/flowers/102/`).

**STL-10.** STL-10 has 10 classes with 500 training images and 800 test images per class [67]. STL-10 is a public dataset but does not specify any particular license (`https://cs.stanford.edu/~acoates/stl10/`).

## C  Results & Insights

### C.1  Additional SHAP Results

We calculate the Pearson correlation metric over all 512 experiments, using the formula described in Sec. A.8. Fig. 8 shows the SHAP Slope plot for the Pearson correlation metric. The gradient-boosting regression model got a $R^2$ of 0.73 when predicting this metric. The variables follow a similar trend as seen in Sec. 4.1.1 for % OOD performance retained metric, where ID class count, augmentation, spatial reduction, and resolution have a positive impact on Pearson correlation. Again, ID class count stands out as the most influential variable for Pearson correlation, suggesting that input diversity enhances OOD generalization. In contrast, depth, over-parameterization level, stem, and CNN vs

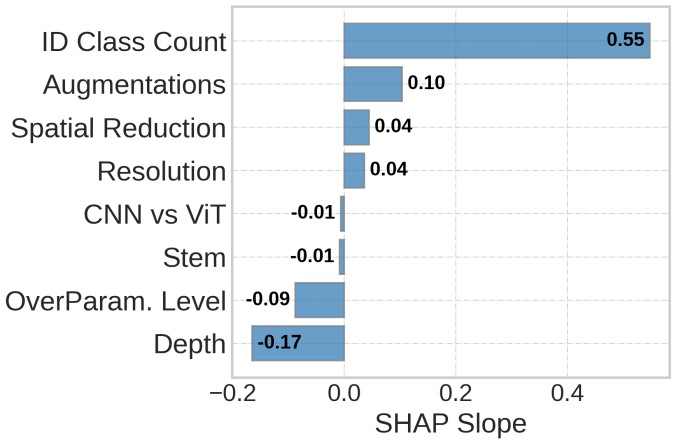

Figure 8: **SHAP results for Pearson correlation.** SHAP slope shows the individual impact of variables on the Pearson correlation target. A positive value indicates enhanced OOD transferability and reduced tunnel effect.

ViT variables negatively impact Pearson correlation. Depth shows the most negative impact among others.

## C.2 Assessing The Tunnel Effect

We conduct 224 experiments on ImageNet-100, exploring various model architectures, and resolutions ($32 \times 32$ and $224 \times 224$), with and without augmentations, and evaluate them on 8 different OOD datasets mentioned in Sec. 3.3. Additionally, to examine resolution in a more fine-grained manner, we conduct 96 experiments with resolutions of $64 \times 64$ and $128 \times 128$ using VGGm-17, ResNet-18, and ViT-T+, also on ImageNet-100 and the same 8 OOD datasets. We conduct 96 more experiments on ImageNet-100, varying class numbers while keeping image numbers constant, and vice versa. To compare the effect of the ID dataset, we perform 64 experiments on CIFAR-10 and CIFAR-100 as ID datasets with their native resolution ($32 \times 32$), exploring VGGm-11 and VGGm-17 models, with and without augmentations, and evaluate them on all OOD datasets. We do the same for VGGm†-11 and VGGm†-17 on CIFAR-100, resulting in another 32 experiments. This totals 512 OOD experiments.

In this section, we summarize results in terms of OOD accuracy and three metrics e.g., % OOD performance retained, Pearson correlation, and ID/OOD alignment. This helps analyze the strength of the tunnel effect and OOD generalization in various transfer settings. The following subsections present results in terms of different metrics and discuss the findings.

### C.2.1 OOD Accuracy

In Table 4, we compare OOD accuracy among models trained on the ImageNet-100 ID dataset with varied resolutions. The insights derived from three metrics regarding the influence of variables on the tunnel effect are consistent with observations of OOD accuracy. Models exhibiting weaker or negligible tunnel effects tend to attain higher OOD accuracy compared to those with stronger tunnel effects.

### C.2.2 % OOD Performance Retained

In Table 5, we compare % OOD performance retained among models trained on the ImageNet-100 ID dataset with varied resolutions. Models exhibiting weaker or no tunnel effect tend to achieve higher % OOD performance retained compared to those displaying a stronger tunnel effect.

Table 4: **OOD Accuracy.** We report average results with 95% confidence intervals (CI). ImageNet-R is abbreviated as IN-R. $\gamma$ denotes over-parameterization level.

| Model | $\gamma$ | Image Size | OOD Accuracy (%) ↑ | | | | | | | | | |
|---|---|---|---|---|---|---|---|---|---|---|---|---|
| | | | NINCO | IN-R | CUB | Aircrafts | Flowers | Pets | CIFAR | STL | Avg. | CI |
| *No Aug* | | | | | | | | | | | | |
| VGGm†-11 | 74.67 | $32^2$ | 54.51 | 20.57 | 19.54 | 16.32 | 43.43 | 47.57 | 49.71 | 75.89 | 40.94 | $24.03 - 57.85$ |
| VGGm-11 | 74.67 | $32^2$ | 51.53 | 15.22 | 17.62 | 16.32 | 40.78 | 43.41 | 39.68 | 68.86 | 36.68 | $21.07 - 52.29$ |
| VGGm-11 | 74.67 | $224^2$ | 59.78 | 19.02 | 17.78 | 17.64 | 41.96 | 38.55 | 63.86 | 77.60 | 42.02 | $23.07 - 60.97$ |
| VGGm†-17 | 158.5 | $32^2$ | 43.20 | 16.32 | 12.94 | 9.24 | 25.98 | 40.50 | 40.94 | 73.47 | 32.82 | $15.51 - 50.13$ |
| VGGm-17 | 158.5 | $32^2$ | 31.80 | 8.43 | 6.94 | 3.99 | 15.00 | 24.21 | 28.07 | 60.42 | 22.36 | $7.27 - 37.45$ |
| VGGm-17 | 158.5 | $224^2$ | 50.51 | 17.52 | 12.15 | 9.51 | 25.98 | 32.30 | 53.15 | 77.94 | 34.88 | $15.44 - 54.32$ |
| ViT-T | 44.28 | $32^2$ | 47.62 | 11.22 | 21.45 | 11.46 | 39.41 | 39.67 | 27.12 | 60.62 | 32.32 | $17.98 - 46.66$ |
| ViT-T | 44.28 | $224^2$ | 54.42 | 11.25 | 16.15 | 9.15 | 26.47 | 33.11 | 39.98 | 65.31 | 31.98 | $15.35 - 48.61$ |
| ViT-T+ | 66.23 | $32^2$ | 46.68 | 10.68 | 19.00 | 10.98 | 35.59 | 37.68 | 24.83 | 59.78 | 30.65 | $16.42 - 44.89$ |
| ViT-T+ | 66.23 | $224^2$ | 50.60 | 8.68 | 11.94 | 7.05 | 21.76 | 30.19 | 33.82 | 59.81 | 27.98 | $12.06 - 43.91$ |
| ResNet-18 | 88.56 | $32^2$ | 24.66 | 5.68 | 4.47 | 4.86 | 9.71 | 20.05 | 16.87 | 48.69 | 16.87 | $4.71 - 29.04$ |
| ResNet-18 | 88.56 | $224^2$ | 54.85 | 18.80 | 21.06 | 17.28 | 43.73 | 38.56 | 53.28 | 74.46 | 40.25 | $23.59 - 56.92$ |
| ResNet-34 | 168.37 | $32^2$ | 24.40 | 4.15 | 4.73 | 4.92 | 9.51 | 18.50 | 13.63 | 45.61 | 15.68 | $4.19 - 27.17$ |
| ResNet-34 | 168.37 | $224^2$ | 52.98 | 15.02 | 17.02 | 15.03 | 32.06 | 33.48 | 49.77 | 71.90 | 35.91 | $18.95 - 52.86$ |
| *Aug* | | | | | | | | | | | | |
| VGGm†-11 | 74.67 | $32^2$ | 65.99 | 22.22 | 32.24 | 21.81 | 66.47 | 55.11 | 56.50 | 80.89 | 50.15 | $32.10 - 68.20$ |
| VGGm-11 | 74.67 | $32^2$ | 62.59 | 18.62 | 26.27 | 14.49 | 56.47 | 47.67 | 49.88 | 76.86 | 44.11 | $25.98 - 62.23$ |
| VGGm-11 | 74.67 | $224^2$ | 70.49 | 24.17 | 33.09 | 28.62 | 70.59 | 50.92 | 68.03 | 85.43 | 53.92 | $35.14 - 72.69$ |
| VGGm†-17 | 158.5 | $32^2$ | 63.10 | 19.15 | 27.87 | 15.39 | 55.78 | 52.18 | 58.11 | 82.18 | 46.72 | $27.57 - 65.87$ |
| VGGm-17 | 158.5 | $32^2$ | 53.06 | 12.13 | 16.34 | 7.98 | 40.88 | 37.01 | 46.55 | 74.39 | 36.04 | $17.45 - 54.64$ |
| VGGm-17 | 158.5 | $224^2$ | 71.68 | 22.62 | 28.65 | 22.59 | 64.12 | 50.45 | 69.80 | 85.39 | 51.91 | $31.84 - 71.98$ |
| ViT-T | 44.28 | $32^2$ | 49.40 | 11.48 | 22.11 | 10.89 | 41.47 | 40.25 | 24.94 | 60.17 | 32.59 | $17.92 - 47.26$ |
| ViT-T | 44.28 | $224^2$ | 66.75 | 17.45 | 35.31 | 17.04 | 60.88 | 43.39 | 51.32 | 74.29 | 45.80 | $28.22 - 63.39$ |
| ViT-T+ | 66.23 | $32^2$ | 54.08 | 13.25 | 24.15 | 11.49 | 42.55 | 44.27 | 30.09 | 64.50 | 35.55 | $20.01 - 51.09$ |
| ViT-T+ | 66.23 | $224^2$ | 72.02 | 18.80 | 37.78 | 19.08 | 64.31 | 47.75 | 61.11 | 79.04 | 49.99 | $31.14 - 68.83$ |
| ResNet-18 | 88.56 | $32^2$ | 51.19 | 13.17 | 19.61 | 11.49 | 39.22 | 41.75 | 36.47 | 65.91 | 34.85 | $19.29 - 50.41$ |
| ResNet-18 | 88.56 | $224^2$ | 70.32 | 27.68 | 34.47 | 21.33 | 62.55 | 57.27 | 68.63 | 84.68 | 53.37 | $34.77 - 71.97$ |
| ResNet-34 | 168.37 | $32^2$ | 49.57 | 12.10 | 18.28 | 10.44 | 37.84 | 39.39 | 35.05 | 66.08 | 33.59 | $17.89 - 49.30$ |
| ResNet-34 | 168.37 | $224^2$ | 70.83 | 23.27 | 30.79 | 22.62 | 58.24 | 52.61 | 66.99 | 83.79 | 51.14 | $32.24 - 70.04$ |

### C.2.3 Pearson Correlation

In Table 6, we compare the Pearson correlation among models trained on the ImageNet-100 ID dataset with varied resolutions. We observe that the Pearson correlation adeptly captures the tunnel effect across different models and settings.

### C.2.4 ID/OOD Alignment

In Table 7, we compare ID/OOD Alignment among models trained on the ImageNet-100 ID dataset with varied resolutions. More performant models obtain higher ID/OOD alignment scores across various experiments compared to less performant models. The impact of different variables and interventions on the ID/OOD alignment is shown in Fig. 19 and Fig. 20.

### C.3 DNN Architecture

In this section, we analyze how DNN architecture impacts % OOD performance retained across transfer settings. To investigate this, We create a boxplot (Fig. 9) that shows % OOD performance retained of each model trained on ImageNet-100 ID dataset with resolutions $32 \times 32$ and $224 \times 224$. We find that DNN architecture greatly impacts the OOD generalization. Among CNNs, VGGm-11 shows superior performance. In terms of CNN vs. ViT, ViTs perform better than CNNs when augmentations are used. Augmentations benefit both CNNs and ViTs, but ViTs seem to benefit more from augmentations.

### C.4 DNN Depth

In this section, we analyze how DNN depth impacts the tunnel effect. Fig. 10 shows that smaller models have higher % OOD performance retained than larger (deeper) models. Therefore, the depth of DNN has an adverse impact on OOD generalization and the tunnel effect.

Table 5: **% OOD Performance Retained.** We report average results with 95% confidence intervals (CI). A higher % OOD performance retained indicates a lesser tunnel effect and better OOD generalization. ImageNet-R is abbreviated as IN-R. $\gamma$ denotes over-parameterization level.

| Model | $\gamma$ | Image Size | % OOD Performance Retained ↑ | | | | | | | | | |
|---|---|---|---|---|---|---|---|---|---|---|---|---|
| | | | NINCO | IN-R | CUB | Aircrafts | Flowers | Pets | CIFAR | STL | Avg. | CI |
| *No Aug* | | | | | | | | | | | | |
| VGGm†-11 | 74.67 | $32^2$ | 88.66 | 94.05 | 74.23 | 81.68 | 72.62 | 100.00 | 89.15 | 97.65 | 87.26 | $78.89 - 95.62$ |
| VGGm-11 | 74.67 | $32^2$ | 88.08 | 75.39 | 65.28 | 59.32 | 69.10 | 94.85 | 84.06 | 92.95 | 78.63 | $67.78 - 89.48$ |
| VGGm-11 | 74.67 | $224^2$ | 89.55 | 87.97 | 75.40 | 82.93 | 64.46 | 100.00 | 87.20 | 98.12 | 85.70 | $76.23 - 95.18$ |
| VGGm†-17 | 158.5 | $32^2$ | 70.36 | 75.08 | 45.90 | 34.57 | 41.67 | 80.80 | 77.19 | 94.21 | 64.97 | $47.49 - 82.45$ |
| VGGm-17 | 158.5 | $32^2$ | 57.45 | 44.58 | 25.94 | 15.15 | 27.67 | 67.06 | 47.69 | 83.65 | 46.15 | $27.44 - 64.86$ |
| VGGm-17 | 158.5 | $224^2$ | 75.19 | 72.78 | 44.22 | 32.02 | 36.20 | 84.27 | 70.23 | 95.73 | 63.83 | $44.72 - 82.94$ |
| ViT-T | 44.28 | $32^2$ | 96.05 | 89.02 | 92.62 | 92.49 | 96.17 | 97.36 | 99.42 | 98.90 | 95.25 | $92.33 - 98.18$ |
| ViT-T | 44.28 | $224^2$ | 83.22 | 60.76 | 43.17 | 42.42 | 40.91 | 74.85 | 76.20 | 86.91 | 63.56 | $47.83 - 79.28$ |
| ViT-T+ | 66.23 | $32^2$ | 95.15 | 87.93 | 87.66 | 87.98 | 90.52 | 91.83 | 97.21 | 97.57 | 91.98 | $88.58 - 95.38$ |
| ViT-T+ | 66.23 | $224^2$ | 76.18 | 50.88 | 32.23 | 32.87 | 34.69 | 65.73 | 68.72 | 81.07 | 55.30 | $38.79 - 71.80$ |
| ResNet-18 | 88.56 | $32^2$ | 49.91 | 36.35 | 19.74 | 22.31 | 23.57 | 58.56 | 44.33 | 75.56 | 41.29 | $25.22 - 57.36$ |
| ResNet-18 | 88.56 | $224^2$ | 86.35 | 72.49 | 74.98 | 66.06 | 77.97 | 92.35 | 80.69 | 95.46 | 80.79 | $72.60 - 88.99$ |
| ResNet-34 | 168.37 | $32^2$ | 51.34 | 28.65 | 20.39 | 22.01 | 21.60 | 48.78 | 42.83 | 72.57 | 38.52 | $23.32 - 53.72$ |
| ResNet-34 | 168.37 | $224^2$ | 79.16 | 60.31 | 51.65 | 40.67 | 48.02 | 83.19 | 65.72 | 90.14 | 64.86 | $50.26 - 79.46$ |
| *Aug* | | | | | | | | | | | | |
| VGGm†-11 | 74.67 | $32^2$ | 100.00 | 100.00 | 99.52 | 96.42 | 96.86 | 100.00 | 99.08 | 100.00 | 98.98 | $97.77 - 100.20$ |
| VGGm-11 | 74.67 | $32^2$ | 98.00 | 90.59 | 81.52 | 56.16 | 86.49 | 97.91 | 87.40 | 97.42 | 86.94 | $75.64 - 98.23$ |
| VGGm-11 | 74.67 | $224^2$ | 96.73 | 100.00 | 95.56 | 99.79 | 92.66 | 100.00 | 96.13 | 100.00 | 97.61 | $95.35 - 99.86$ |
| VGGm†-17 | 158.5 | $32^2$ | 92.98 | 84.24 | 83.98 | 56.81 | 82.94 | 100.00 | 91.13 | 99.43 | 86.44 | $75.25 - 97.63$ |
| VGGm-17 | 158.5 | $32^2$ | 83.53 | 55.70 | 48.92 | 31.89 | 61.96 | 87.63 | 65.06 | 93.28 | 66.00 | $48.84 - 83.15$ |
| VGGm-17 | 158.5 | $224^2$ | 98.02 | 87.66 | 80.94 | 71.37 | 83.31 | 100.00 | 91.74 | 99.42 | 89.06 | $80.74 - 97.37$ |
| ViT-T | 44.28 | $32^2$ | 97.48 | 97.18 | 94.26 | 90.75 | 91.76 | 97.86 | 98.82 | 98.87 | 95.87 | $93.26 - 98.48$ |
| ViT-T | 44.28 | $224^2$ | 95.50 | 93.23 | 89.38 | 87.25 | 90.52 | 95.88 | 96.19 | 98.23 | 93.27 | $90.13 - 96.41$ |
| ViT-T+ | 66.23 | $32^2$ | 97.55 | 96.36 | 91.02 | 88.05 | 88.75 | 97.44 | 98.14 | 99.14 | 94.56 | $90.87 - 98.24$ |
| ViT-T+ | 66.23 | $224^2$ | 98.37 | 89.52 | 87.49 | 87.97 | 89.13 | 97.35 | 95.56 | 99.03 | 93.05 | $88.99 - 97.11$ |
| ResNet-18 | 88.56 | $32^2$ | 83.03 | 63.15 | 58.89 | 50.80 | 69.20 | 85.28 | 74.14 | 90.17 | 71.83 | $60.55 - 83.11$ |
| ResNet-18 | 88.56 | $224^2$ | 93.03 | 85.84 | 83.10 | 72.85 | 85.64 | 95.85 | 91.49 | 97.82 | 88.20 | $81.60 - 94.80$ |
| ResNet-34 | 168.37 | $32^2$ | 80.97 | 57.26 | 56.63 | 43.02 | 68.08 | 80.93 | 71.63 | 89.90 | 68.55 | $55.87 - 81.24$ |
| ResNet-34 | 168.37 | $224^2$ | 91.24 | 71.19 | 68.69 | 55.16 | 74.16 | 91.99 | 84.84 | 96.40 | 79.21 | $67.62 - 90.80$ |

## C.5 Overparameterization Level

In this section, we investigate the relationship between overparameterization level and % OOD performance retained. We use experimental results from models trained on ImageNet-100 ID dataset with resolutions $32 \times 32$ and $224 \times 224$ for this analysis. To do this, we compute average % OOD performance retained across overparameterization levels and create a boxplot using this data. Fig. 11 shows that % OOD performance retained decreases as over-parameterization level increases. Therefore, overparameterization level has an adverse impact on OOD generalization.

## C.6 Stem

This section analyzes how the stem affects tunnel formation. We use experimental results of models trained on ImageNet-100 ID dataset with resolutions $32 \times 32$ and $224 \times 224$. For CNNs, the stem represents kernel size ($3 \times 3$ and $7 \times 7$) at the stem layer whereas, for ViTs, it is patch size ($8 \times 8$). Fig. 12 reveals that increasing stem hurts % OOD performance retained. In particular, when comparing CNNs (VGGm's $3 \times 3$ and ResNet's $7 \times 7$), increasing stem from 3 to 7 decreases average % OOD performance retention. However, ViTs ($8 \times 8$) perform better than CNNs. On the other hand, stem significantly impacts ID/OOD alignment. Across all compared models, increasing stem shows a significant negative impact on ID/OOD alignment.

## C.7 Augmentation

In this section, we study how augmentation affects the tunnel effect. We used all 512 experiments to assess the impact of augmentation. 256 experiments used augmentations and another 256 experiments did not use any augmentation. In experiments with augmentation, we used random resized crop and random horizontal flip. Fig. 13 exhibits that increasing augmentation greatly improves % OOD performance retained.

Table 6: **Pearson Correlation.** We report average results with 95% confidence intervals (CI). A higher Pearson correlation indicates a lesser tunnel effect and better OOD generalization. ImageNet-R is abbreviated as IN-R. $\gamma$ denotes over-parameterization level.

| Model | $\gamma$ | Image Size | Pearson Correlation ↑ | | | | | | | | | |
|---|---|---|---|---|---|---|---|---|---|---|---|---|
| | | | NINCO | IN-R | CUB | Aircrafts | Flowers | Pets | CIFAR | STL | Avg. | CI |
| *No Aug* | | | | | | | | | | | | |
| VGGm†-11 | 74.67 | $32^2$ | 0.93 | 0.98 | 0.88 | 0.91 | 0.87 | 0.98 | 0.93 | 0.97 | 0.93 | $0.90 - 0.97$ |
| VGGm-11 | 74.67 | $32^2$ | 0.96 | 0.93 | 0.85 | 0.85 | 0.87 | 0.98 | 0.92 | 0.95 | 0.91 | $0.87 - 0.96$ |
| VGGm-11 | 74.67 | $224^2$ | 0.92 | 0.97 | 0.86 | 0.88 | 0.81 | 0.99 | 0.92 | 0.96 | 0.91 | $0.86 - 0.96$ |
| VGGm†-17 | 158.5 | $32^2$ | 0.84 | 0.94 | 0.76 | 0.71 | 0.70 | 0.96 | 0.87 | 0.95 | 0.84 | $0.75 - 0.93$ |
| VGGm-17 | 158.5 | $32^2$ | 0.62 | 0.61 | 0.38 | 0.38 | 0.35 | 0.88 | 0.55 | 0.86 | 0.58 | $0.41 - 0.75$ |
| VGGm-17 | 158.5 | $224^2$ | 0.84 | 0.92 | 0.69 | 0.66 | 0.62 | 0.97 | 0.84 | 0.95 | 0.81 | $0.70 - 0.92$ |
| ViT-T | 44.28 | $32^2$ | 0.94 | 0.97 | 0.97 | 0.96 | 0.94 | 0.99 | 0.97 | 0.96 | 0.96 | $0.95 - 0.98$ |
| ViT-T | 44.28 | $224^2$ | 0.75 | 0.70 | 0.53 | 0.55 | 0.40 | 0.90 | 0.61 | 0.83 | 0.66 | $0.52 - 0.79$ |
| ViT-T+ | 66.23 | $32^2$ | 0.95 | 0.95 | 0.93 | 0.87 | 0.88 | 0.97 | 0.97 | 0.95 | 0.94 | $0.90 - 0.97$ |
| ViT-T+ | 66.23 | $224^2$ | 0.68 | 0.59 | 0.46 | 0.47 | 0.42 | 0.83 | 0.62 | 0.74 | 0.60 | $0.48 - 0.72$ |
| ResNet-18 | 88.56 | $32^2$ | 0.83 | 0.76 | 0.69 | 0.71 | 0.68 | 0.90 | 0.77 | 0.92 | 0.78 | $0.71 - 0.86$ |
| ResNet-18 | 88.56 | $224^2$ | 0.97 | 0.97 | 0.96 | 0.93 | 0.97 | 0.98 | 0.96 | 0.99 | 0.96 | $0.95 - 0.98$ |
| ResNet-34 | 168.37 | $32^2$ | 0.78 | 0.68 | 0.58 | 0.62 | 0.48 | 0.83 | 0.72 | 0.87 | 0.69 | $0.59 - 0.80$ |
| ResNet-34 | 168.37 | $224^2$ | 0.87 | 0.90 | 0.84 | 0.81 | 0.74 | 0.98 | 0.79 | 0.91 | 0.86 | $0.79 - 0.92$ |
| *Aug* | | | | | | | | | | | | |
| VGGm†-11 | 74.67 | $32^2$ | 0.95 | 0.99 | 0.98 | 0.95 | 0.94 | 0.99 | 0.95 | 0.95 | 0.96 | $0.95 - 0.98$ |
| VGGm-11 | 74.67 | $32^2$ | 0.97 | 0.95 | 0.94 | 0.78 | 0.93 | 0.99 | 0.92 | 0.97 | 0.93 | $0.88 - 0.98$ |
| VGGm-11 | 74.67 | $224^2$ | 0.96 | 0.99 | 0.98 | 0.95 | 0.95 | 0.99 | 0.97 | 0.97 | 0.97 | $0.96 - 0.98$ |
| VGGm†-17 | 158.5 | $32^2$ | 0.94 | 0.96 | 0.94 | 0.78 | 0.90 | 0.99 | 0.94 | 0.97 | 0.93 | $0.87 - 0.98$ |
| VGGm-17 | 158.5 | $32^2$ | 0.85 | 0.62 | 0.60 | 0.30 | 0.64 | 0.97 | 0.62 | 0.93 | 0.69 | $0.51 - 0.87$ |
| VGGm-17 | 158.5 | $224^2$ | 0.95 | 0.97 | 0.94 | 0.86 | 0.91 | 0.99 | 0.95 | 0.96 | 0.94 | $0.91 - 0.97$ |
| ViT-T | 44.28 | $32^2$ | 0.96 | 0.99 | 0.99 | 0.95 | 0.97 | 0.99 | 0.99 | 0.99 | 0.98 | $0.96 - 0.99$ |
| ViT-T | 44.28 | $224^2$ | 0.95 | 0.99 | 0.98 | 0.98 | 0.98 | 0.99 | 0.95 | 0.98 | 0.97 | $0.96 - 0.99$ |
| ViT-T+ | 66.23 | $32^2$ | 0.96 | 0.99 | 0.98 | 0.96 | 0.97 | 1.00 | 0.99 | 0.99 | 0.98 | $0.97 - 0.99$ |
| ViT-T+ | 66.23 | $224^2$ | 0.97 | 0.98 | 0.98 | 0.98 | 0.96 | 0.99 | 0.94 | 0.98 | 0.97 | $0.96 - 0.99$ |
| ResNet-18 | 88.56 | $32^2$ | 0.96 | 0.90 | 0.91 | 0.83 | 0.92 | 0.98 | 0.91 | 0.96 | 0.92 | $0.88 - 0.96$ |
| ResNet-18 | 88.56 | $224^2$ | 0.98 | 0.98 | 0.97 | 0.95 | 0.99 | 0.97 | 0.99 | 0.99 | 0.98 | $0.96 - 0.99$ |
| ResNet-34 | 168.37 | $32^2$ | 0.90 | 0.81 | 0.79 | 0.41 | 0.77 | 0.98 | 0.82 | 0.92 | 0.80 | $0.66 - 0.94$ |
| ResNet-34 | 168.37 | $224^2$ | 0.93 | 0.96 | 0.95 | 0.84 | 0.87 | 0.99 | 0.94 | 0.94 | 0.93 | $0.89 - 0.97$ |

## C.8 ID Class Count

This section explores the relationship between the ID class count and the tunnel effect. We use experimental results from the VGGm-11 models trained with ImageNet-100, CIFAR-10, and CIFAR-100 using $32 \times 32$ resolution and image augmentations. Fig. 14 displays how the number of ID classes affects the tunnel effect and OOD performance. We see that more ID classes reduce the strength of the tunnel effect and enhance OOD generalization.

For the statistical analysis, we consider 16 paired experiments (32 total) comparing CIFAR-10 and CIFAR-100 datasets (ID). The compared models are VGGm-11, VGGm-17, VGGm†-11, and VGGm†-17. Increasing the number of CIFAR classes from 10 to 100 increased % OOD performance retained from 35.81% to 73.42% ($p < 0.001$), with a *large* effect size ($|\delta| = 0.758$). Similarly, the Pearson correlation was improved from 0.54 to 0.88 ($p < 0.001$), with a *large* effect size ($|\delta| = 0.742$). And, ID/OOD alignment reached 0.22 from 0.12 ($p = 0.013$), with a *large* effect size ($|\delta| = 0.531$).

## C.9 Number of ID Training Classes vs. ID Dataset Size

Fig. 15 shows OOD performance for various dataset configurations with varied classes and samples per class. Increasing class counts significantly impacts OOD performance, whereas, increasing the number of training samples has less impact.

## C.10 The Tunnel Effect Is Not Specific To CIFAR-10

In Fig. 16 and Fig. 14, we observe that the tunnel effect is present in various ID datasets e.g., ImageNet-100, CIFAR-10, and CIFAR-100. Therefore, the tunnel effect is not a characteristic of a particular ID dataset such as CIFAR-10.

Table 7: **ID/OOD Alignment.** We report average results with 95% confidence intervals (CI). A higher ID/OOD alignment indicates a lesser tunnel effect and better OOD generalization. ImageNet-R is abbreviated as IN-R. $\gamma$ denotes over-parameterization level.

| Model | $\gamma$ | Image Size | ID/OOD Alignment ↑ | | | | | | | | | |
|---|---|---|---|---|---|---|---|---|---|---|---|---|
| | | | NINCO | IN-R | CUB | Aircrafts | Flowers | Pets | CIFAR | STL | Avg. | CI |
| *No Aug* | | | | | | | | | | | | |
| VGGm†-11 | 74.67 | $32^2$ | 0.32 | 0.12 | 0.12 | 0.09 | 0.26 | 0.29 | 0.28 | 0.40 | 0.23 | $0.14 - 0.33$ |
| VGGm-11 | 74.67 | $32^2$ | 0.25 | 0.07 | 0.09 | 0.08 | 0.20 | 0.19 | 0.21 | 0.30 | 0.17 | $0.10 - 0.24$ |
| VGGm-11 | 74.67 | $224^2$ | 0.45 | 0.14 | 0.13 | 0.13 | 0.31 | 0.47 | 0.29 | 0.52 | 0.30 | $0.17 - 0.44$ |
| VGGm†-17 | 158.5 | $32^2$ | 0.27 | 0.10 | 0.08 | 0.05 | 0.16 | 0.24 | 0.25 | 0.40 | 0.19 | $0.10 - 0.29$ |
| VGGm-17 | 158.5 | $32^2$ | 0.15 | 0.04 | 0.03 | 0.02 | 0.07 | 0.13 | 0.12 | 0.26 | 0.10 | $0.04 - 0.17$ |
| VGGm-17 | 158.5 | $224^2$ | 0.39 | 0.13 | 0.09 | 0.07 | 0.20 | 0.40 | 0.25 | 0.54 | 0.26 | $0.12 - 0.39$ |
| ViT-T | 44.28 | $32^2$ | 0.16 | 0.04 | 0.07 | 0.04 | 0.14 | 0.09 | 0.14 | 0.18 | 0.11 | $0.06 - 0.15$ |
| ViT-T | 44.28 | $224^2$ | 0.33 | 0.07 | 0.10 | 0.05 | 0.16 | 0.23 | 0.20 | 0.34 | 0.18 | $0.09 - 0.27$ |
| ViT-T+ | 66.23 | $32^2$ | 0.16 | 0.04 | 0.06 | 0.03 | 0.12 | 0.08 | 0.13 | 0.17 | 0.10 | $0.06 - 0.14$ |
| ViT-T+ | 66.23 | $224^2$ | 0.29 | 0.05 | 0.07 | 0.04 | 0.12 | 0.18 | 0.17 | 0.29 | 0.15 | $0.07 - 0.23$ |
| ResNet-18 | 88.56 | $32^2$ | 0.08 | 0.02 | 0.01 | 0.01 | 0.03 | 0.05 | 0.06 | 0.13 | 0.05 | $0.02 - 0.08$ |
| ResNet-18 | 88.56 | $224^2$ | 0.36 | 0.12 | 0.14 | 0.11 | 0.29 | 0.34 | 0.25 | 0.43 | 0.26 | $0.16 - 0.35$ |
| ResNet-34 | 168.37 | $32^2$ | 0.07 | 0.01 | 0.01 | 0.01 | 0.03 | 0.03 | 0.05 | 0.10 | 0.04 | $0.01 - 0.07$ |
| ResNet-34 | 168.37 | $224^2$ | 0.34 | 0.10 | 0.11 | 0.09 | 0.21 | 0.31 | 0.22 | 0.41 | 0.22 | $0.12 - 0.32$ |
| *Aug* | | | | | | | | | | | | |
| VGGm†-11 | 74.67 | $32^2$ | 0.44 | 0.15 | 0.22 | 0.14 | 0.44 | 0.37 | 0.37 | 0.48 | 0.32 | $0.21 - 0.44$ |
| VGGm-11 | 74.67 | $32^2$ | 0.38 | 0.11 | 0.16 | 0.08 | 0.35 | 0.29 | 0.29 | 0.42 | 0.26 | $0.16 - 0.36$ |
| VGGm-11 | 74.67 | $224^2$ | 0.55 | 0.19 | 0.26 | 0.22 | 0.55 | 0.52 | 0.40 | 0.60 | 0.41 | $0.27 - 0.55$ |
| VGGm†-17 | 158.5 | $32^2$ | 0.44 | 0.13 | 0.20 | 0.10 | 0.39 | 0.40 | 0.37 | 0.52 | 0.32 | $0.19 - 0.44$ |
| VGGm-17 | 158.5 | $32^2$ | 0.33 | 0.08 | 0.10 | 0.05 | 0.26 | 0.28 | 0.23 | 0.42 | 0.22 | $0.11 - 0.33$ |
| VGGm-17 | 158.5 | $224^2$ | 0.59 | 0.19 | 0.24 | 0.18 | 0.53 | 0.57 | 0.42 | 0.64 | 0.42 | $0.26 - 0.58$ |
| ViT-T | 44.28 | $32^2$ | 0.17 | 0.04 | 0.07 | 0.03 | 0.14 | 0.08 | 0.14 | 0.17 | 0.10 | $0.06 - 0.15$ |
| ViT-T | 44.28 | $224^2$ | 0.39 | 0.10 | 0.21 | 0.10 | 0.36 | 0.29 | 0.26 | 0.39 | 0.26 | $0.17 - 0.36$ |
| ViT-T+ | 66.23 | $32^2$ | 0.21 | 0.05 | 0.09 | 0.04 | 0.17 | 0.11 | 0.17 | 0.22 | 0.13 | $0.08 - 0.19$ |
| ViT-T+ | 66.23 | $224^2$ | 0.49 | 0.13 | 0.26 | 0.13 | 0.44 | 0.40 | 0.32 | 0.48 | 0.33 | $0.21 - 0.45$ |
| ResNet-18 | 88.56 | $32^2$ | 0.24 | 0.06 | 0.09 | 0.05 | 0.19 | 0.16 | 0.20 | 0.27 | 0.16 | $0.09 - 0.23$ |
| ResNet-18 | 88.56 | $224^2$ | 0.55 | 0.22 | 0.27 | 0.16 | 0.50 | 0.53 | 0.45 | 0.60 | 0.41 | $0.27 - 0.55$ |
| ResNet-34 | 168.37 | $32^2$ | 0.23 | 0.05 | 0.08 | 0.04 | 0.17 | 0.15 | 0.18 | 0.26 | 0.15 | $0.08 - 0.21$ |
| ResNet-34 | 168.37 | $224^2$ | 0.56 | 0.18 | 0.25 | 0.18 | 0.46 | 0.52 | 0.42 | 0.60 | 0.40 | $0.26 - 0.54$ |

Table 8: **OOD Accuracy of Large Pre-trained Models.** We report average results with 95% confidence intervals (CI). ImageNet-R is abbreviated as IN-R. $\gamma$ denotes over-parameterization level.

| Model | $\gamma$ | OOD Accuracy (%) ↑ | | | | | | | | | |
|---|---|---|---|---|---|---|---|---|---|---|---|
| | | NINCO | IN-R | CUB | Aircrafts | Flowers | Pets | CIFAR | STL | Avg. | CI |
| *SSL Models* | | | | | | | | | | | |
| SwAV ResNet-50 | 19.98 | 80.78 | 35.87 | 43.84 | 18.39 | 50.88 | 67.84 | 77.98 | 94.91 | 58.81 | $37.73 - 79.90$ |
| DINO ViT-B | 66.97 | 88.78 | 50.00 | 55.02 | 37.95 | 87.06 | 80.29 | 86.75 | 96.61 | 72.81 | $55.04 - 90.58$ |
| MAE ViT-B | 66.97 | 84.27 | 43.35 | 31.26 | 27.27 | 77.84 | 68.31 | 79.04 | 95.21 | 63.32 | $42.27 - 84.37$ |
| MUGS ViT-B | 66.97 | 94.22 | 64.53 | 81.74 | 53.26 | 92.25 | 85.83 | 93.92 | 99.00 | 83.09 | $69.94 - 96.25$ |
| ConvNextV2 | 69.47 | 91.24 | 67.48 | 65.33 | 26.55 | 54.90 | 83.26 | 91.52 | 98.70 | 72.37 | $52.85 - 91.89$ |
| *SL Models* | | | | | | | | | | | |
| ResNet-50 | 19.98 | 88.10 | 47.03 | 45.69 | 17.22 | 60.00 | 66.01 | 89.86 | 97.28 | 63.90 | $41.70 - 86.09$ |
| ViT-B | 66.97 | 88.52 | 52.48 | 48.43 | 29.52 | 83.73 | 80.59 | 86.18 | 96.85 | 70.79 | $51.22 - 90.35$ |
| ConvNextV1 | 69.47 | 90.31 | 65.95 | 63.48 | 30.24 | 53.04 | 81.55 | 91.63 | 98.18 | 71.80 | $53.03 - 90.56$ |

## C.11 Large Pre-trained Models Do Not Exhibit The Tunnel Effect

With the prevalence of large pre-trained models, many research and applications are driven by these models. They show impressive transferability, especially those trained with SSL methods. Our goal is to explain this phenomenon using the tunnel effect hypothesis. Although the tunnel effect is hypothesized to be present in all over-parameterized DNNs, in [9], only supervised DNNs were studied. Using pre-trained models, we study both SSL and SL models. For SSL pre-trained models, we study the following:

- ResNet-50 pre-trained with SwAV [73],
- ViT-B pre-trained with DINO V1 [75],
- ViT-B pre-trained with MAE [50],
- ViT-B pre-trained with MUGS [74], and

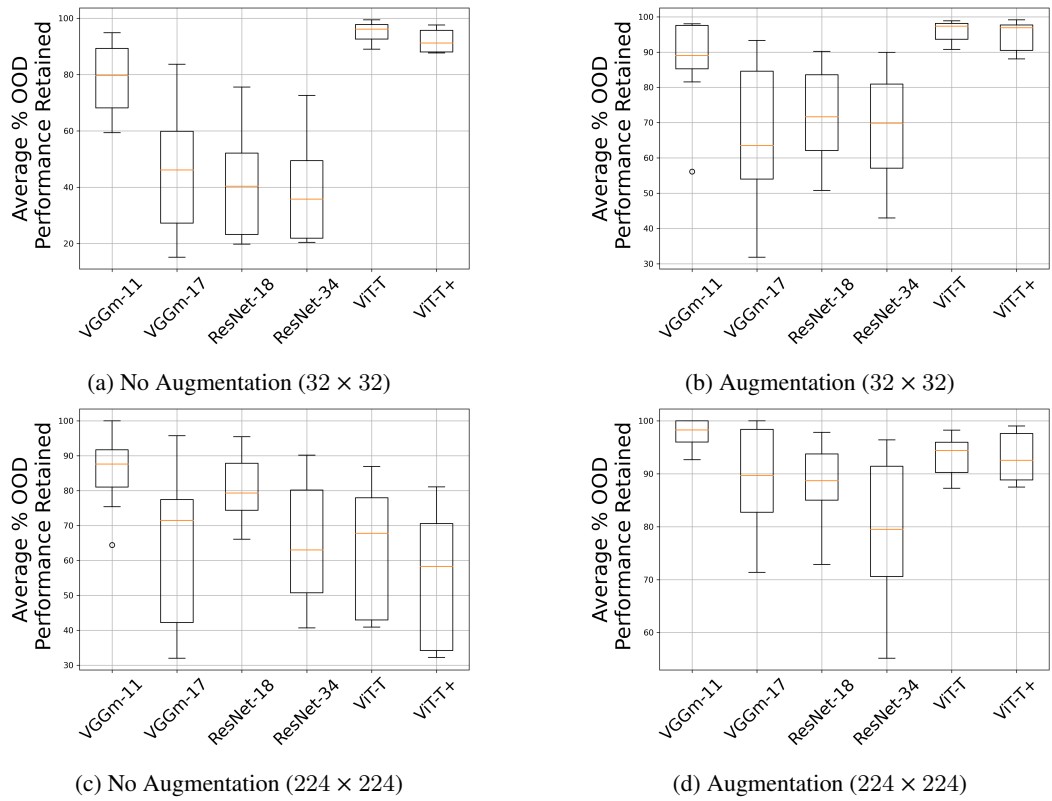

(a) No Augmentation ($32 \times 32$)

(b) Augmentation ($32 \times 32$)

(c) No Augmentation ($224 \times 224$)

(d) Augmentation ($224 \times 224$)

Figure 9: **Box-plots of % OOD Performance Retained.** The figure shows the % OOD performance retained, computed across models trained on ImageNet-100 at resolutions $32 \times 32$ and $224 \times 224$, both with and without augmentation.

Table 9: **% OOD Performance Retained of Large Pre-trained Models.** We report average results with 95% confidence intervals (CI). ImageNet-R is abbreviated as IN-R. $\gamma$ denotes over-parameterization level.

| Model | $\gamma$ | % OOD Performance Retained ↑ | | | | | | | | | CI |
|---|---|---|---|---|---|---|---|---|---|---|---|
| | | **NINCO** | **IN-R** | **CUB** | **Aircrafts** | **Flowers** | **Pets** | **CIFAR** | **STL** | **Avg.** | |
| *SSL Models* | | | | | | | | | | | |
| SwAV ResNet-50 | 19.98 | 95.77 | 83.12 | 87.83 | 58.10 | 69.85 | 94.91 | 100.00 | 100.00 | 86.20 | $73.80 - 98.59$ |
| DINO ViT-B | 66.97 | 100.00 | 100.00 | 100.00 | 100.00 | 98.45 | 99.48 | 100.00 | 100.00 | 99.74 | $99.29 - 100.19$ |
| MAE ViT-B | 66.97 | 98.90 | 98.52 | 91.28 | 100.00 | 96.24 | 98.61 | 100.00 | 100.00 | 97.94 | $95.52 - 100.36$ |
| MUGS ViT-B | 66.97 | 99.91 | 100.00 | 100.00 | 95.89 | 99.79 | 100.00 | 100.00 | 100.00 | 99.45 | $98.27 - 100.62$ |
| ConvNextV2 | 69.47 | 97.90 | 99.39 | 92.66 | 84.61 | 89.31 | 99.49 | 99.70 | 99.96 | 95.38 | $90.61 - 100.14$ |
| *SL Models* | | | | | | | | | | | |
| ResNet-50 | 19.98 | 96.73 | 92.01 | 73.88 | 37.30 | 65.67 | 91.12 | 99.49 | 99.82 | 82.00 | $64.15 - 99.85$ |
| ViT-B | 66.97 | 99.43 | 95.60 | 94.80 | 87.31 | 93.74 | 98.96 | 100.00 | 99.60 | 96.18 | $92.65 - 99.71$ |
| ConvNextV1 | 69.47 | 98.52 | 98.19 | 92.71 | 90.00 | 89.27 | 98.87 | 99.56 | 99.67 | 95.85 | $92.24 - 99.46$ |

- ConvNeXt-B V2 pre-trained with FCMAE [76].

For SL pre-trained models, we study ResNet-50 [54], ConvNext-B V1 [72], and ViT-B [55] which are pre-trained with SL loss and data labels. We carry out **72 experiments** in total of which 27 are for 3 SL models and 45 are for 5 SSL models. All SL and SSL models are pre-trained and fine-tuned (SSL only) on ImageNet-1K dataset. In total, we trained **1980 linear probes** for ID and OOD datasets. Appendix A.6 gives additional details and model configurations.

Due to limited computational resources, we could not pre-train our own SSL models to do a rigorous paired analysis. SSL training is approximately 16× more expensive than SL, considering multiplicative factors, e.g., ~ 8× more epochs and ~ 2× more forward passes. Thus, we could only evaluate publicly available pre-trained models released by their creators. We find that SHAP analysis appears to be less informative due to the small sample size and because most of the pre-trained models do

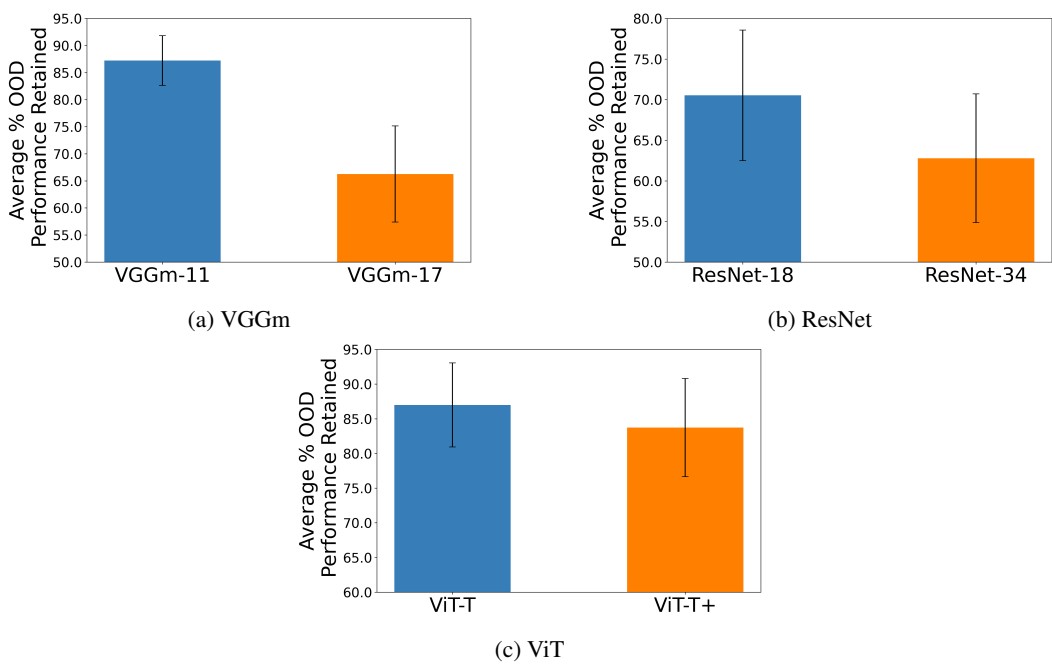

Figure 10: **Impact of Depth.** The figure shows how depth impacts the tunnel effect in terms of % OOD performance retained. Increasing depth decreases the OOD performance retention and intensifies the tunnel effect. The Y-axis shows the average with a 95% confidence interval.

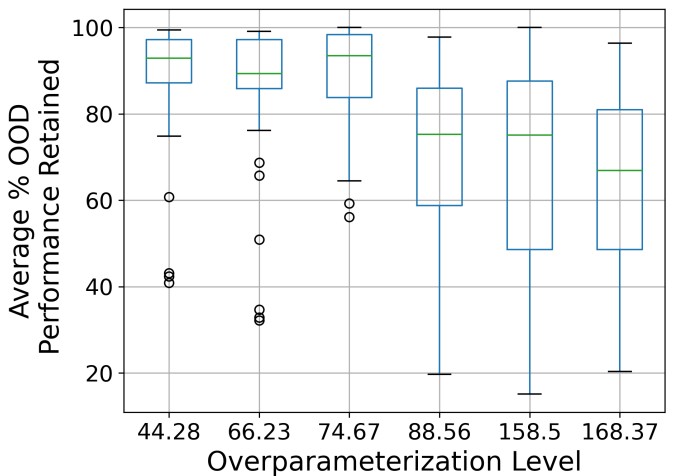

Figure 11: **Over-parameterization level.** This figure exhibits % OOD performance retained computed across over-parameterization levels. This is based on models trained on ImagerNet-100 at resolutions $32 \times 32$ and $224 \times 224$. Increasing the over-parameterization level reduces the % OOD performance retained and intensifies the tunnel effect.

not exhibit the tunnel effect, resulting in less variability. Therefore, we focus on OOD linear probe analysis.

In Fig. 17 and Table 9, we observe that the tunnel is not present in most cases except for the ResNet-50 models. As shown in Table 8, the SSL method, MUGS achieves the highest average OOD accuracy among all methods. In terms of DNN architecture, ViT-B exhibits better OOD performance than ResNet-50. Interestingly, among CNNs, ConvNeXt-B, which adapts design choices from Swin-

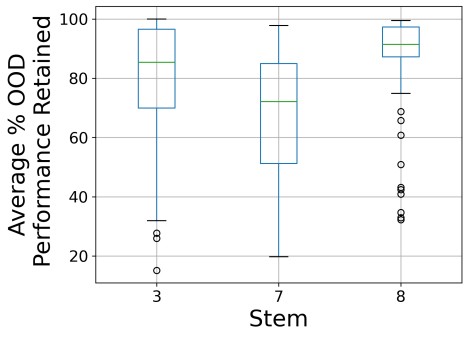
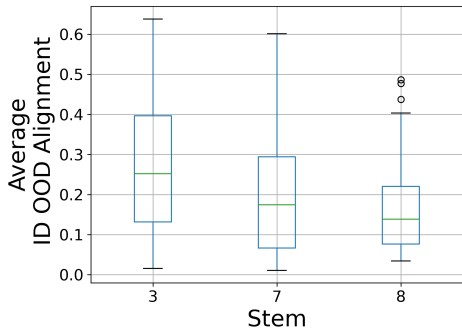

(a) % OOD Performance Retained

(b) ID/OOD Alignment

Figure 12: **Stem.** This figure exhibits % OOD performance retained and ID/OOD alignment computed across stems and averaged across OOD datasets. This is based on models trained on ImagerNet-100 (ID) with $32 \times 32$ and $224 \times 224$ resolutions. When comparing CNNs (VGGm's $3 \times 3$ and ResNet's $7 \times 7$), increasing stem exhibits a negative impact on % OOD performance retained. Across all models, increasing stem negatively impacts ID/OOD alignment.

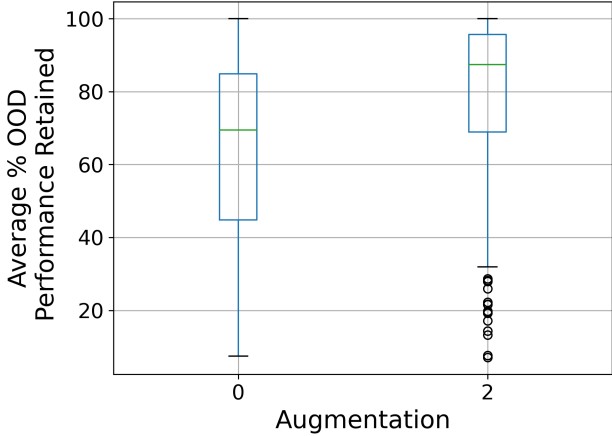

Figure 13: **Augmentation.** This figure exhibits % OOD performance retained to assess the impact of augmentation. The comparison is made between 256 experiments without augmentation and 256 experiments with augmentations. Augmentation greatly enhances the % OOD performance retained and reduces the tunnel effect.

transformer [95], shows better OOD performance than ResNet-50. They also rival ViTs. This corroborates our argument that DNN architecture matters for OOD performance.

The comparison between SSL and SL models is not straightforward as there are multiple factors involved, e.g., pretext tasks, training epochs, augmentations, and so on. Except for MAE, other SSL methods exhibit strong OOD performance compared to SL counterparts (Table 8). Previous works [52, 96] also found that MAE performs poorly in linear evaluation. When we compared SL vs. SSL models in the same family, in terms of average OOD accuracy, SSL ($71.42\%$) outperforms SL ($68.83\%$). As illustrated in Fig. 18, SSL models perform better than SL models in terms of % OOD performance retained. We report the ID accuracy of these models in Table 10. Given that SSL methods employ more augmentations than their SL counterparts (see Table 3), the performance improvements of SSL methods over SL ones are likely, in part, due to stronger augmentations, which is also argued in prior studies [97, 98]. While complimenting earlier works, our work confirms that large pre-trained models do not form tunnels, adding valuable perspectives to current knowledge.

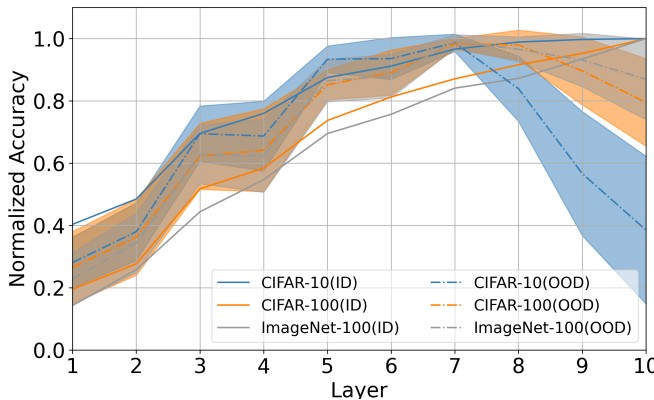

Figure 14: **ID class count.** This figure compares the OOD and ID linear probe accuracy for VGGm-11 models across three ID datasets ($32 \times 32$ resolution): CIFAR-10, CIFAR-100, and ImageNet-100. The OOD curve is the average of 8 OOD datasets and the shaded area denotes standard deviation. The strength of the tunnel is weaker for ImageNet-100 and CIFAR-100 compared to CIFAR-10.

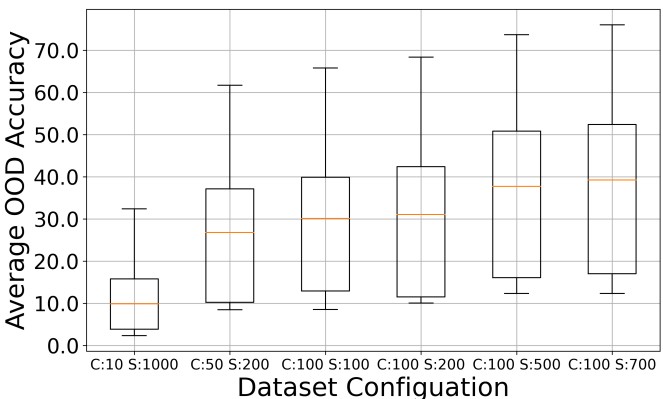

Figure 15: **Class Count vs. Data Quantity.** The figure shows the trend that models trained with more classes and more samples have better OOD accuracy (%). We use C and S to denote the number of classes and number of samples per class respectively, in the X-axis of the figure.

## C.12 Examining Actual Accuracy

We also examine the impact of different variables and interventions on the ID performance, OOD performance, and ID/OOD alignment by analyzing the actual accuracy (no normalization). We show these results in Fig. 19 and Fig. 20. We observe that the actual accuracy trends are consistent with our previous findings where variables that improve (degrade) OOD generalization also show higher (lower) ID/OOD alignment.

## C.13 Additional Resolution Results

Here we present additional resolution results. The image resolution shows a significant impact on the tunnel effect and OOD transferability. As shown in Fig. 19a, higher resolution images achieve higher performance in terms of ID accuracy, OOD accuracy, and ID/OOD alignment.

**CNN.** Fig. 21 and Fig. 22 show the results for VGGm-17 models trained on ImageNet-100 with varied image resolutions in experiments with and without augmentations. The tunnel effect decreases with the increase in image resolution in all experiments.

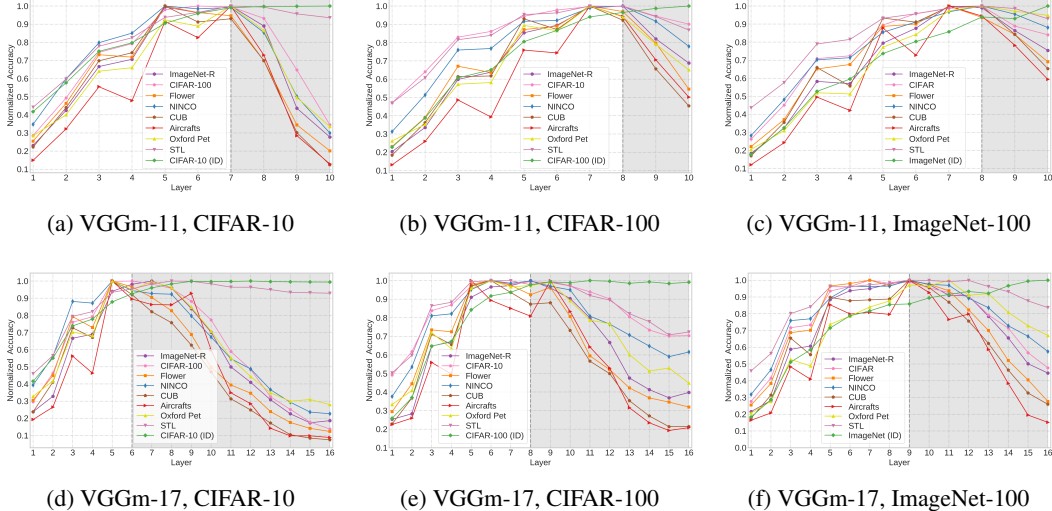

(a) VGGm-11, CIFAR-10          (b) VGGm-11, CIFAR-100          (c) VGGm-11, ImageNet-100

(d) VGGm-17, CIFAR-10          (e) VGGm-17, CIFAR-100          (f) VGGm-17, ImageNet-100

Figure 16: **The tunnel effect is not specific to a particular ID dataset.** The OOD linear probe accuracy for VGGm models trained on various ID datasets: CIFAR-10, CIFAR-100, and ImageNet-100 with low resolution ($32 \times 32$) images in augmentation-free settings. The gray shaded area denotes the tunnel. The tunnel effect is prominent in all low-resolution settings.

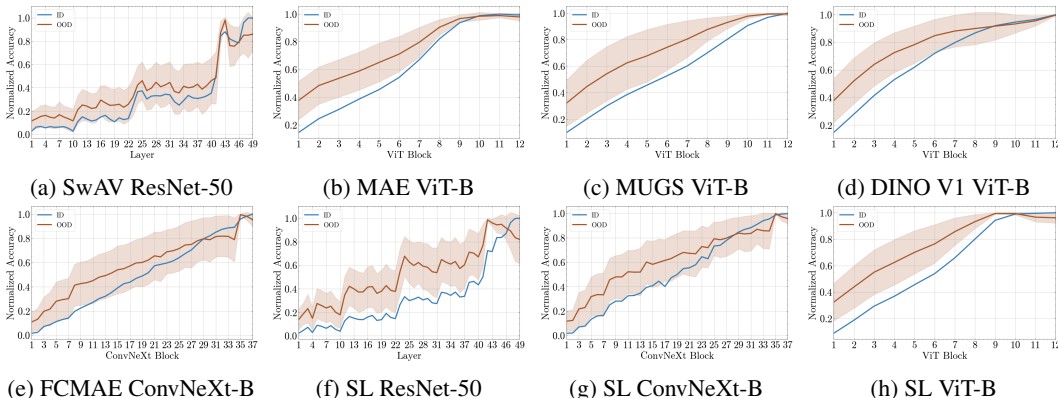

(a) SwAV ResNet-50     (b) MAE ViT-B     (c) MUGS ViT-B     (d) DINO V1 ViT-B

(e) FCMAE ConvNeXt-B   (f) SL ResNet-50  (g) SL ConvNeXt-B  (h) SL ViT-B

Figure 17: **OOD performance of large pre-trained models.** We show normalized linear probe accuracy on the ID (ImageNet-1K) and OOD datasets. The OOD curve is the average of 8 OOD datasets. Shaded area denotes the standard deviation. Most models do not show any tunnel effect.

**ViT.** Fig. 23 and Fig. 24 show the results for ViT-T+ models trained on ImageNet-100 with varied image resolutions in experiments with and without augmentations. When augmentation is omitted, low-resolution inputs show a weaker tunnel effect than high-resolution inputs. However, both low- and high-resolution inputs significantly mitigate the tunnel effect when augmentations are used.

**Rank Analysis.** We examine the numerical rank of the representations evaluated on the ID dataset, following prior work [9]. In Fig. 25, we observe that a model trained on high-resolution images maintains a higher rank than a model trained on low-resolution images, corroborating our previous findings (Sec. 4.1.3). The representations rank for resolution $32 \times 32$ plummets after the extractor, exhibiting the neural collapse phenomenon [10] whereas resolution $224 \times 224$ retains a much higher rank in the corresponding layers. The degradation in OOD accuracy and ID representation rank is more pronounced for low-resolution images than for high-resolution images. As a result, a model trained on low-resolution images exhibits a stronger tunnel effect than one trained on high-resolution images.

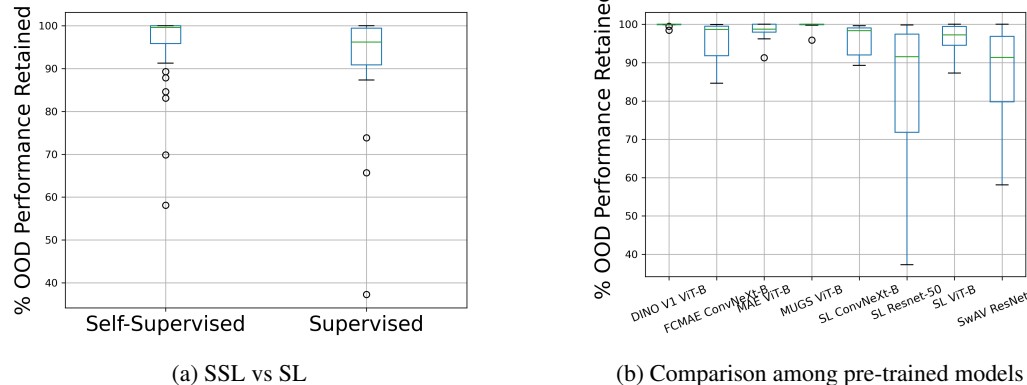

(a) SSL vs SL

(b) Comparison among pre-trained models

Figure 18: **Summary plots for large pre-trained models.** The left figure exhibits % OOD performance retained computed across training methods (SL and SSL). The right figure displays % OOD performance retained computed across distinct architectures. SSL models show higher % OOD performance retained than SL models.

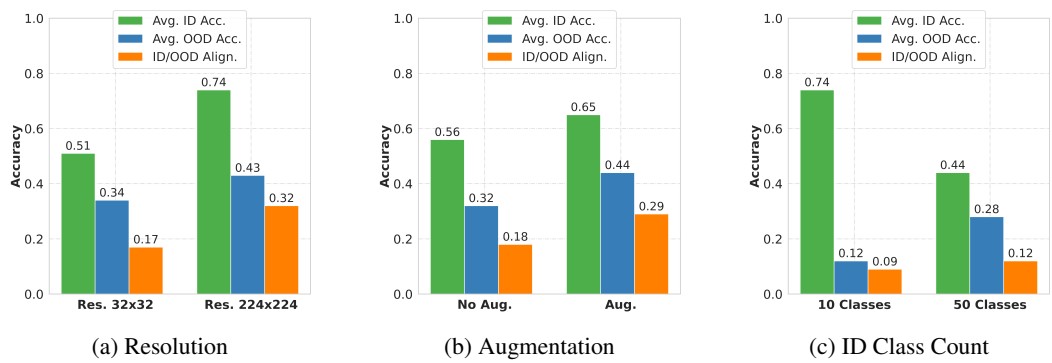

(a) Resolution

(b) Augmentation

(c) ID Class Count

Figure 19: **Dataset variables**. Impact of resolution, augmentation, and ID class count on ID accuracy, OOD accuracy, and ID/OOD alignment. This analysis is based on actual accuracy (no normalization) averaged over models and datasets. The accuracy and alignment score range from 0 to 1.

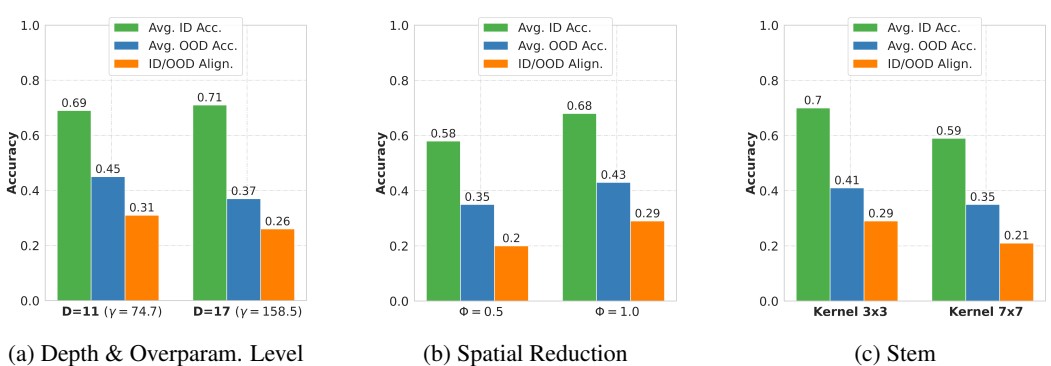

(a) Depth & Overparam. Level

(b) Spatial Reduction

(c) Stem

Figure 20: **DNN architecture variables**. Impact of depth (D), overparameterization level ($\gamma$), spatial reduction ($\phi$), and stem on ID accuracy, OOD accuracy, and ID/OOD alignment. This analysis is based on actual accuracy (no normalization) averaged over models and datasets. The accuracy and alignment score range from 0 to 1.

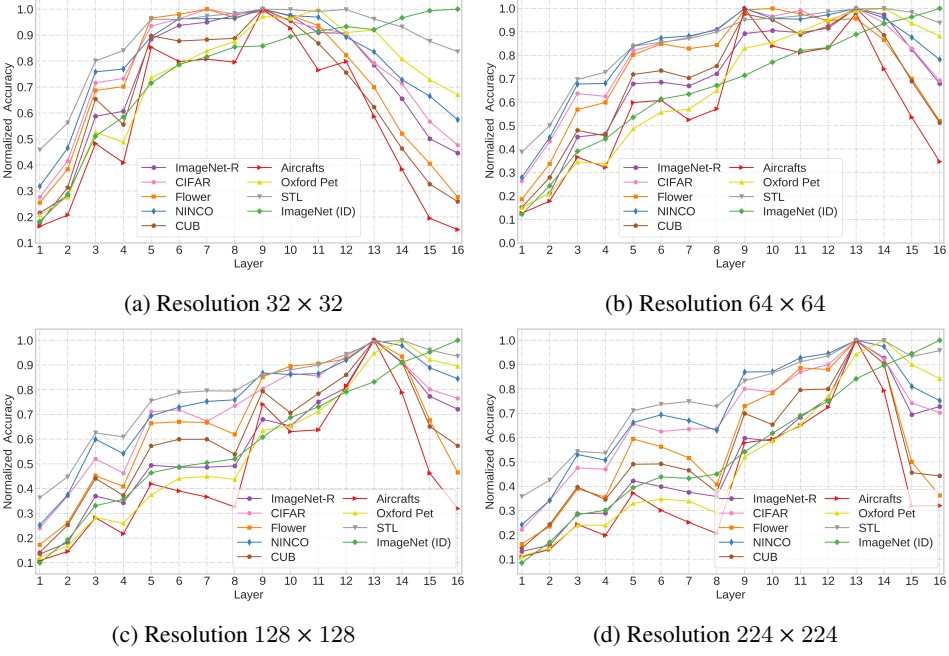

(a) Resolution $32 \times 32$

(b) Resolution $64 \times 64$

(c) Resolution $128 \times 128$

(d) Resolution $224 \times 224$

Figure 21: The OOD linear probe accuracy for similar sized **VGGm-17** models trained on ImageNet-100 dataset with **varied image resolutions in augmentation-free settings**. The OOD degradation reduces with the increase in image resolution.

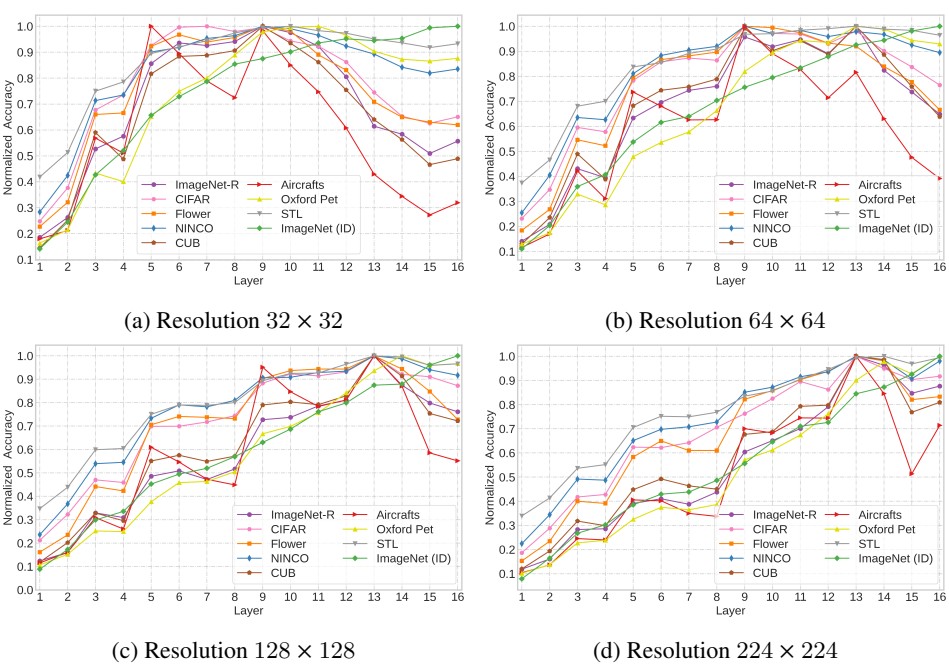

(a) Resolution $32 \times 32$

(b) Resolution $64 \times 64$

(c) Resolution $128 \times 128$

(d) Resolution $224 \times 224$

Figure 22: The OOD linear probe accuracy for similar sized **VGGm-17** models trained on ImageNet-100 dataset with **varied image resolutions while using image augmentations**. The OOD degradation reduces with the increase in image resolution.

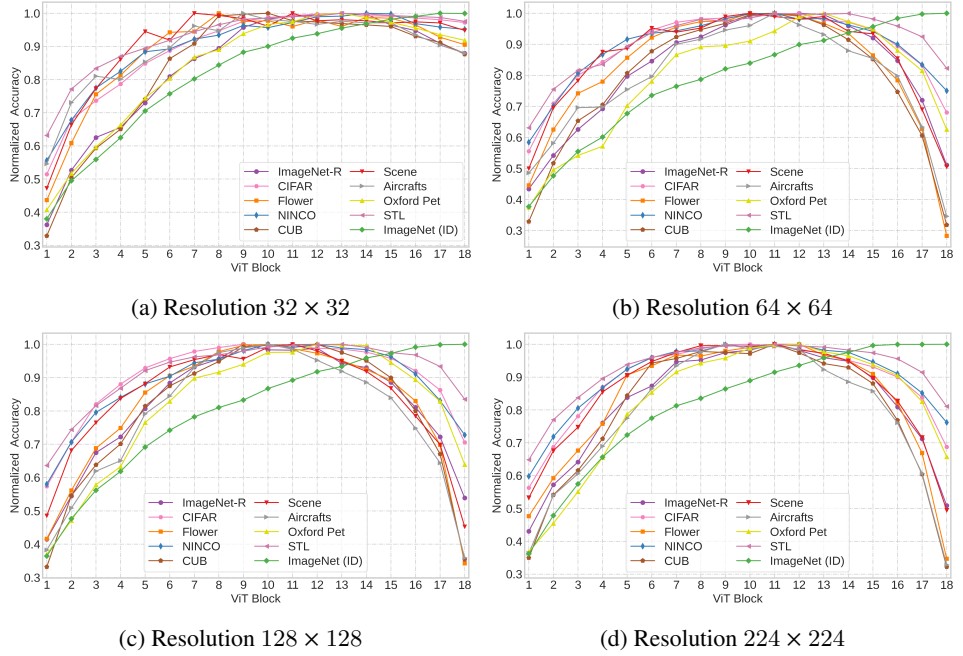

(a) Resolution $32 \times 32$

(b) Resolution $64 \times 64$

(c) Resolution $128 \times 128$

(d) Resolution $224 \times 224$

Figure 23: The OOD linear probe accuracy for similar sized **ViT-T+** models trained on ImageNet-100 dataset with **varied image resolutions in augmentation-free settings**. The OOD degradation reduces with the increase in image resolution.

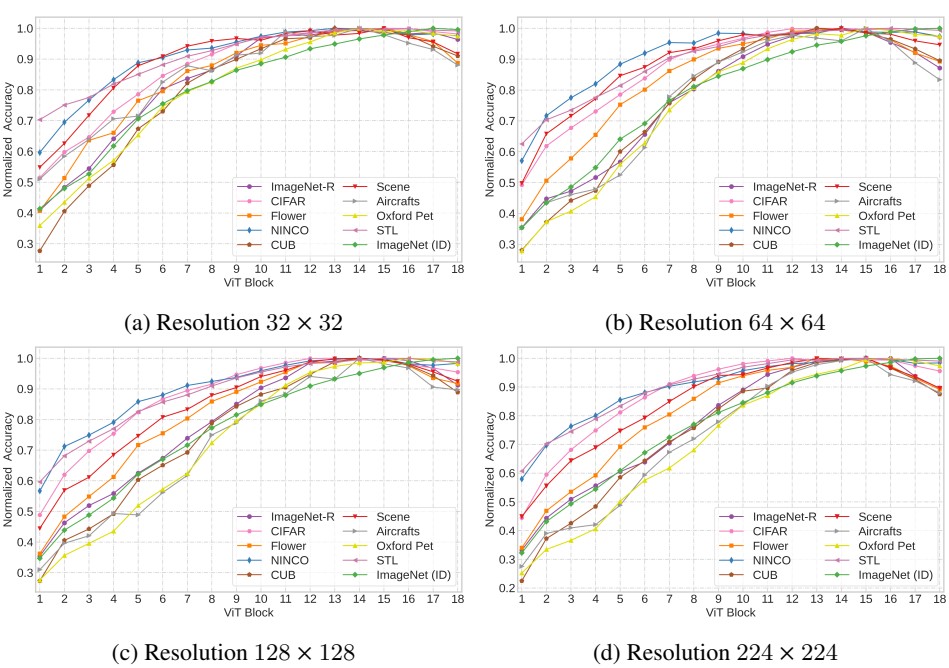

(a) Resolution $32 \times 32$

(b) Resolution $64 \times 64$

(c) Resolution $128 \times 128$

(d) Resolution $224 \times 224$

Figure 24: The OOD linear probe accuracy for similar sized **ViT-T+** models trained on ImageNet-100 dataset with **varied image resolutions while using image augmentations**. The OOD degradation reduces with the increase in image resolution.

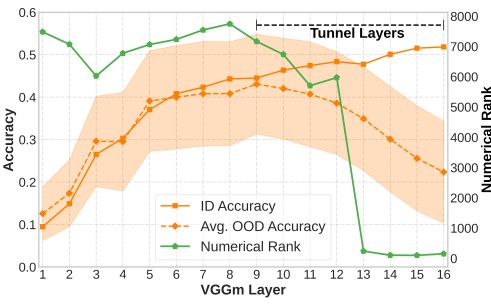
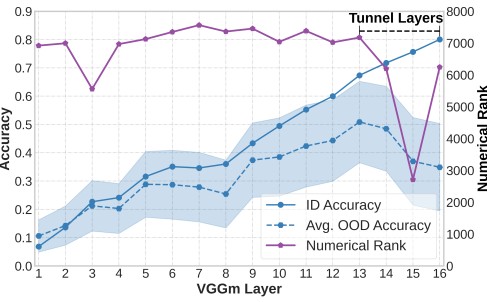

(a) Resolution $32 \times 32$ (strong tunnel effect)    (b) Resolution $224 \times 224$ (weak tunnel effect)

Figure 25: **High-resolution model maintains a high representation rank**. This figure corresponds to Fig. 1 in the main text. In this setting, identical VGGm-17 architectures are trained on identical ID datasets (100 ImageNet classes), where only the resolution is changed. The Y-axis shows the actual accuracy (no normalization) of linear probes trained on ID and OOD datasets. The OOD curve is the average of 8 OOD datasets (Sec. 3.3), with the standard deviation denoted with shading. The Y-axis also shows the numerical rank evaluated on the ID dataset. Here the OOD accuracy and ID representation rank align with the dynamics of the tunnel effect: the stronger the tunnel effect, the lower the OOD accuracy and ID representation rank.

## D In-Distribution Performance

In this section, we present the ID performance of all pre-trained DNNs. Table 10 shows the ID performance of 8 large-scale off-the-shelf models pre-tained on ImageNet-1K dataset.

Table 10: **ID performance of large pre-trained models.** Reported is the best top-1 accuracy (%) on ImageNet-1K dataset ($224 \times 224$).

| Model | Accuracy |
|---|---|
| **SSL** | |
| SwAV ResNet-50 | 75.3 |
| DINO V1 ViT-B | 78.2 |
| MAE ViT-B | 83.6 |
| MUGS ViT-B | 80.6 |
| FCMAE ConvNeXtV2 | 84.9 |
| **SL** | |
| ResNet-50 | 80.9 |
| ViT-B | 81.1 |
| ConvNeXtV1 | 83.8 |

Table 11 presents the results for each of the **64 DNN models** studied in our main results. For each model and ID dataset combination, it gives the ID accuracy for the resolution/augmentation combination.

On **ImageNet-100**, for ViT-T+, ResNet-18, and VGGm-17, 8 models were trained per DNN architecture (4 resolution x 2 augmentation policies). The total is 24 DNNs. For the same ID dataset, 4 models were trained per DNN architecture, namely, ViT-T, ResNet-34, and VGGm-11 (2 resolution $\times$ 2 augmentation policies). The total is 12 DNNs. An additional 2 models were trained per architecture for VGGm†-11 and VGGm†-17 (1 resolution $\times$ 2 augmentation policies). The total is 4 DNNs. Overall, *40 DNNs* were trained for ImageNet-100.

On **CIFAR-100**, for VGGm-11, VGGm†-11, VGGm17, and VGGm†-17, 2 models were trained per architecture (1 resolution $\times$ 2 augmentation policies). The total is *8 DNNs*.

On **CIFAR-10**, for VGGm-11, VGGm-17, 2 models were trained per architecture (1 resolution $\times$ 2 augmentation polices). The total is *4 DNNs*.

On **ImageNet Subsets**, for VGGm-11, 12 models were trained (1 resolution $times$ 6 dataset configurations $\times$ 2 augmentation policies). The total is *12 DNNs*.

Table 11: **ID performance of DNNs trained on CIFAR-10, CIFAR-100, ImageNet-100, and ImageNet Subsets.** Reported is the best top-1 accuracy (%) for experiments with varied resolution and augmentation policies. The total number of DNN parameters is in Million and denoted by $\#P$. For each ID dataset, we group results based on DNN architecture. The sum of rows indicates the total number of DNNs i.e., 64.

| ID Dataset | Model | #P | Resolution | No Aug ↑ | Aug ↑ |
|---|---|---|---|---|---|
| **ImageNet-100** | ViT-T | 5.61 | $32^2$ | 36.48 | 35.28 |
| | | | $224^2$ | 62.50 | 60.72 |
| | ViT-T+ | 8.39 | $32^2$ | 35.64 | 41.14 |
| | | | $64^2$ | 47.18 | 56.64 |
| | | | $128^2$ | 56.84 | 66.60 |
| | | | $224^2$ | 59.38 | 70.24 |
| | ResNet-18 | 11.22 | $32^2$ | 34.38 | 49.70 |
| | | | $64^2$ | 45.56 | 66.08 |
| | | | $128^2$ | 59.90 | 76.94 |
| | | | $224^2$ | 68.80 | 81.74 |
| | ResNet-34 | 21.33 | $32^2$ | 32.10 | 50.80 |
| | | | $224^2$ | 68.86 | 83.12 |
| | VGGm-11 | 9.46 | $32^2$ | 52.22 | 63.74 |
| | | | $224^2$ | 78.74 | 83.02 |
| | VGGm-17 | 20.08 | $32^2$ | 52.02 | 65.78 |
| | | | $64^2$ | 66.10 | 77.88 |
| | | | $128^2$ | 75.62 | 84.76 |
| | | | $224^2$ | 80.28 | 86.28 |
| | VGGm†-11 | 9.46 | $32^2$ | 63.12 | 70.28 |
| | VGGm†-17 | 20.08 | $32^2$ | 65.76 | 74.40 |
| **CIFAR-100** | VGGm-11 | 9.46 | $32^2$ | 63.78 | 72.36 |
| | VGGm-17 | 20.08 | $32^2$ | 60.91 | 70.92 |
| | VGGm†-11 | 9.46 | $32^2$ | 71.18 | 76.00 |
| | VGGm†-17 | 20.08 | $32^2$ | 70.23 | 76.19 |
| **CIFAR-10** | VGGm-11 | 9.46 | $32^2$ | 88.05 | 93.04 |
| | VGGm-17 | 20.08 | $32^2$ | 88.20 | 92.69 |
| **ImageNet Subsets** 10 classes & 1K samples | VGGm-11 | 9.46 | $32^2$ | 70.2 | 77.0 |
| 50 classes & 200 samples | | | | 37.2 | 50.7 |
| 100 classes & 100 samples | | | | 26.1 | 33.2 |
| 100 classes & 200 samples | | | | 32.8 | 45.0 |
| 100 classes & 500 samples | | | | 44.7 | 57.1 |
| 100 classes & 700 samples | | | | 47.5 | 61.6 |

# E    Additional Statistical Results

In this section, we report additional statistical results for different variables and interventions. Table 12 includes average results with confidence interval. Table 13 reports effect size (Cliff's Delta) and $p$-value (Wilcoxon signed-rank test) for various pair comparisons where $\#E$ denotes the number of paired experiments (total number of experiments is $2 \times E$).

Table 12: **Average Results.** We report average results in 3 metrics with a 95% confidence interval (CI) for different variables and interventions.

| Variable | % OOD Performance Retained ↑ | | Pearson Correlation ↑ | | ID/OOD Alignment ↑ | |
|---|---|---|---|---|---|---|
| | Avg. | CI | Avg. | CI | Avg. | CI |
| *Resolution* | | | | | | |
| $32^2$ | 68.63 | $61.18 - 76.09$ | 0.81 | $0.76 - 0.87$ | 0.13 | $0.10 - 0.15$ |
| $64^2$ | 69.78 | $63.48 - 76.08$ | 0.83 | $0.78 - 0.88$ | 0.20 | $0.16 - 0.24$ |
| $128^2$ | 77.25 | $71.81 - 82.68$ | 0.88 | $0.83 - 0.92$ | 0.28 | $0.23 - 0.32$ |
| $224^2$ | 78.37 | $72.66 - 84.08$ | 0.88 | $0.83 - 0.92$ | 0.30 | $0.25 - 0.35$ |
| *Augmentation* | | | | | | |
| No Aug. | 64.26 | $61.20 - 67.31$ | 0.77 | $0.74 - 0.80$ | 0.15 | $0.14 - 0.17$ |
| Aug. | 78.41 | $75.65 - 81.16$ | 0.86 | $0.84 - 0.89$ | 0.25 | $0.23 - 0.27$ |
| *Stem* | | | | | | |
| 3 | 76.74 | $71.21 - 82.27$ | 0.84 | $0.80 - 0.89$ | 0.27 | $0.23 - 0.31$ |
| 7 | 66.66 | $61.09 - 72.23$ | 0.87 | $0.83 - 0.90$ | 0.21 | $0.17 - 0.25$ |
| 8 | 85.35 | $80.81 - 89.90$ | 0.88 | $0.84 - 0.92$ | 0.17 | $0.14 - 0.20$ |
| *DNN Arch.* | | | | | | |
| VGGm | 75.38 | $71.10 - 79.66$ | 0.85 | $0.82 - 0.88$ | 0.27 | $0.24 - 0.31$ |
| ResNet | 68.97 | $64.64 - 73.30$ | 0.89 | $0.87 - 0.91$ | 0.22 | $0.19 - 0.25$ |
| ViT | 81.66 | $77.45 - 85.88$ | 0.85 | $0.80 - 0.89$ | 0.18 | $0.15 - 0.20$ |
| *Spatial Reduction* | | | | | | |
| 1 | 84.40 | $80.25 - 88.54$ | 0.92 | $0.90 - 0.94$ | 0.26 | $0.23 - 0.30$ |
| 0.5 | 64.85 | $59.06 - 70.63$ | 0.72 | $0.66 - 0.78$ | 0.18 | $0.15 - 0.21$ |
| *Depth* | | | | | | |
| 11 | 89.19 | $85.71 - 92.66$ | 0.94 | $0.92 - 0.95$ | 0.28 | $0.24 - 0.33$ |
| 12 | 86.99 | $80.94 - 93.04$ | 0.89 | $0.84 - 0.95$ | 0.16 | $0.13 - 0.20$ |
| 17 | 69.41 | $62.56 - 76.25$ | 0.80 | $0.74 - 0.85$ | 0.25 | $0.20 - 0.30$ |
| $18(ResNet)$ | 70.53 | $62.50 - 78.56$ | 0.91 | $0.88 - 0.94$ | 0.22 | $0.16 - 0.28$ |
| $18(ViT)$ | 83.72 | $76.65 - 90.79$ | 0.87 | $0.81 - 0.94$ | 0.18 | $0.13 - 0.22$ |
| 34 | 62.78 | $54.88 - 70.69$ | 0.82 | $0.77 - 0.87$ | 0.20 | $0.14 - 0.26$ |
| *Overparam Level* | | | | | | |
| 44.28 | 86.99 | $80.94 - 93.04$ | 0.89 | $0.84 - 0.95$ | 0.16 | $0.13 - 0.20$ |
| 66.23 | 83.72 | $76.65 - 90.79$ | 0.87 | $0.81 - 0.94$ | 0.18 | $0.13 - 0.22$ |
| 74.67 | 87.22 | $82.62 - 91.82$ | 0.93 | $0.91 - 0.95$ | 0.29 | $0.23 - 0.34$ |
| 88.56 | 70.53 | $62.50 - 78.56$ | 0.91 | $0.88 - 0.94$ | 0.22 | $0.16 - 0.28$ |
| 158.5 | 66.26 | $57.39 - 75.13$ | 0.76 | $0.68 - 0.83$ | 0.25 | $0.18 - 0.32$ |
| 168.37 | 62.78 | $54.88 - 70.69$ | 0.82 | $0.77 - 0.87$ | 0.20 | $0.14 - 0.26$ |
| *ID Dataset* | | | | | | |
| ImageNet-100 | 82.78 | $75.47 - 90.10$ | 0.92 | $0.89 - 0.95$ | 0.22 | $0.16 - 0.28$ |
| CIFAR-10 | 35.81 | $22.60 - 49.02$ | 0.54 | $0.42 - 0.65$ | 0.12 | $0.04 - 0.20$ |
| CIFAR-100 | 73.42 | $64.62 - 82.22$ | 0.88 | $0.84 - 0.93$ | 0.22 | $0.15 - 0.29$ |
| *Classes_Samples* | | | | | | |
| C10_S1000 | 29.61 | $24.01 - 35.21$ | 0.42 | $0.37 - 0.47$ | 0.06 | $0.03 - 0.08$ |
| C50_S200 | 68.72 | $61.34 - 76.11$ | 0.84 | $0.79 - 0.88$ | 0.11 | $0.07 - 0.15$ |
| C100_S100 | 79.57 | $72.81 - 86.33$ | 0.91 | $0.87 - 0.94$ | 0.08 | $0.05 - 0.11$ |
| C100_S200 | 74.28 | $66.78 - 81.78$ | 0.89 | $0.85 - 0.93$ | 0.11 | $0.07 - 0.15$ |
| C100_S500 | 78.04 | $69.83 - 86.24$ | 0.90 | $0.86 - 0.94$ | 0.17 | $0.12 - 0.23$ |
| C100_S700 | 77.19 | $68.20 - 86.18$ | 0.91 | $0.87 - 0.95$ | 0.19 | $0.13 - 0.25$ |

Table 13: **Effect Size & *p*-value.** A bigger effect size, $|\delta|$ indicates a bigger statistical difference between each pair and the order is negligible $(N) < $ small $(S) < $ medium $(M) < $ large $(L)$.

| Pair | % OOD Perf. Retained | | Pearson Corr. | | ID/OOD Alignment | | #E |
|------|---------|---------|---------|---------|---------|---------|-----|
| | $|\delta| \uparrow$ | $p$−value ↓ | $|\delta| \uparrow$ | $p$−value ↓ | $|\delta| \uparrow$ | $p$−value ↓ | |
| *Aug.* Aug. vs No Aug. | $0.370(M)$ | $< 0.001$ | $0.374(M)$ | $< 0.001$ | $0.357(M)$ | $< 0.001$ | 256 |
| *Spatial Red.* 1 vs 0.5 | $0.531(L)$ | $< 0.001$ | $0.536(L)$ | $< 0.001$ | $0.361(M)$ | $< 0.001$ | 64 |
| *Resolution* | | | | | | | |
| $32^2$ vs $64^2$ | $0.002(N)$ | $0.315$ | $0.023(N)$ | $0.572$ | $0.326(S)$ | $< 0.001$ | 48 |
| $32^2$ vs $128^2$ | $0.171(S)$ | $0.001$ | $0.240(S)$ | $0.006$ | $0.567(L)$ | $< 0.001$ | 48 |
| $32^2$ vs $224^2$ | $0.198(S)$ | $0.005$ | $0.280(S)$ | $0.011$ | $0.625(L)$ | $< 0.001$ | 48 |
| $64^2$ vs $128^2$ | $0.208(S)$ | $< 0.001$ | $0.222(S)$ | $< 0.001$ | $0.283(S)$ | $< 0.001$ | 48 |
| $64^2$ vs $224^2$ | $0.250(S)$ | $< 0.001$ | $0.251(S)$ | $0.077$ | $0.351(M)$ | $< 0.001$ | 48 |
| $128^2$ vs $224^2$ | $0.059(N)$ | $0.741$ | $0.042(N)$ | $0.947$ | $0.102(N)$ | $0.035$ | 48 |
| *Stem* | | | | | | | |
| 3 vs 7 | $0.306(S)$ | $< 0.001$ | $0.004(N)$ | $0.145$ | $0.226(S)$ | $< 0.001$ | 64 |
| 3 vs 8 | $0.249(S)$ | $0.014$ | $0.293(S)$ | $0.092$ | $0.346(M)$ | $< 0.001$ | 64 |
| 7 vs 8 | $0.580(L)$ | $< 0.001$ | $0.250(S)$ | $0.138$ | $0.083(N)$ | $< 0.001$ | 64 |
| *Depth* | | | | | | | |
| 11 vs 12 | $0.041(N)$ | $0.978$ | $0.150(S)$ | $0.963$ | $0.475(L)$ | $< 0.001$ | 32 |
| 11 vs 17 | $0.539(L)$ | $< 0.001$ | $0.497(L)$ | $< 0.001$ | $0.161(S)$ | $< 0.001$ | 48 |
| 11 vs 18$(ResNet)$ | $0.520(L)$ | $< 0.001$ | $0.025(N)$ | $0.322$ | $0.285(S)$ | $< 0.001$ | 32 |
| 11 vs 18$(ViT)$ | $0.051(N)$ | $0.512$ | $0.031(N)$ | $0.846$ | $0.443(M)$ | $< 0.001$ | 32 |
| 11 vs 34 | $0.680(L)$ | $< 0.001$ | $0.570(L)$ | $< 0.001$ | $0.340(M)$ | $< 0.001$ | 32 |
| 12 vs 17 | $0.534(L)$ | $< 0.001$ | $0.533(L)$ | $0.013$ | $0.250(S)$ | $< 0.001$ | 32 |
| 12 vs 18$(ResNet)$ | $0.574(L)$ | $0.003$ | $0.123(N)$ | $0.625$ | $0.125(N)$ | $0.002$ | 32 |
| 12 vs 18$(ViT)$ | $0.128(N)$ | $< 0.001$ | $0.088(N)$ | $0.010$ | $0.020(N)$ | $0.190$ | 32 |
| 12 vs 34 | $0.689(L)$ | $< 0.001$ | $0.471(M)$ | $0.052$ | $0.070(N)$ | $0.024$ | 32 |
| 17 vs 18$(ResNet)$ | $0.090(N)$ | $0.262$ | $0.516(L)$ | $< 0.001$ | $0.100(N)$ | $0.003$ | 32 |
| 17 vs 18$(ViT)$ | $0.472(M)$ | $< 0.001$ | $0.457(M)$ | $0.025$ | $0.217(S)$ | $< 0.001$ | 32 |
| 17 vs 34 | $0.113(N)$ | $0.062$ | $0.096(N)$ | $0.005$ | $0.178(S)$ | $< 0.001$ | 32 |
| 18$(Res.)$ vs 18$(ViT)$ | $0.465(M)$ | $0.017$ | $0.021(N)$ | $0.818$ | $0.102(N)$ | $0.004$ | 32 |
| 18$(ResNet)$ vs 34 | $0.242(S)$ | $< 0.001$ | $0.443(M)$ | $< 0.001$ | $0.072(N)$ | $< 0.001$ | 32 |
| 18$(ViT)$ vs 34 | $0.590(L)$ | $< 0.001$ | $0.387(M)$ | $0.106$ | $0.037(N)$ | $0.045$ | 32 |
| *Overparam Level* | | | | | | | |
| 44.28 vs 66.23 | $0.128(N)$ | $< 0.001$ | $0.088(N)$ | $0.010$ | $0.020(N)$ | $0.190$ | 32 |
| 44.28 vs 74.67 | $0.041(N)$ | $0.978$ | $0.150(S)$ | $0.963$ | $0.475(L)$ | $< 0.001$ | 32 |
| 44.28 vs 88.56 | $0.574(L)$ | $0.003$ | $0.123(N)$ | $0.625$ | $0.125(N)$ | $0.002$ | 32 |
| 44.28 vs 158.5 | $0.534(L)$ | $< 0.001$ | $0.533(L)$ | $0.013$ | $0.250(S)$ | $< 0.001$ | 32 |
| 44.28 vs 168.37 | $0.689(L)$ | $< 0.001$ | $0.471(M)$ | $0.052$ | $0.070(N)$ | $0.024$ | 32 |
| 66.23 vs 74.67 | $0.051(N)$ | $0.512$ | $0.031(N)$ | $0.846$ | $0.443(M)$ | $< 0.001$ | 32 |
| 66.23 vs 88.56 | $0.465(M)$ | $0.017$ | $0.021(N)$ | $0.818$ | $0.102(N)$ | $0.004$ | 32 |
| 66.23 vs 158.5 | $0.472(M)$ | $< 0.001$ | $0.457(M)$ | $0.025$ | $0.217(S)$ | $< 0.001$ | 32 |
| 66.23 vs 168.37 | $0.590(L)$ | $< 0.001$ | $0.387(M)$ | $0.106$ | $0.037(N)$ | $0.045$ | 32 |
| 74.67 vs 88.56 | $0.520(L)$ | $< 0.001$ | $0.025(N)$ | $0.322$ | $0.285(S)$ | $< 0.001$ | 32 |
| 74.67 vs 158.5 | $0.531(L)$ | $< 0.001$ | $0.553(L)$ | $< 0.001$ | $0.174(S)$ | $< 0.001$ | 32 |
| 74.67 vs 168.37 | $0.680(L)$ | $< 0.001$ | $0.570(L)$ | $< 0.001$ | $0.340(M)$ | $< 0.001$ | 32 |
| 88.56 vs 158.5 | $0.090(N)$ | $0.262$ | $0.516(L)$ | $< 0.001$ | $0.100(N)$ | $0.003$ | 32 |
| 88.56 vs 168.37 | $0.242(S)$ | $< 0.001$ | $0.443(M)$ | $< 0.001$ | $0.072(N)$ | $< 0.001$ | 32 |
| 158.5 vs 168.37 | $0.113(N)$ | $0.062$ | $0.096(N)$ | $0.005$ | $0.178(S)$ | $< 0.001$ | 32 |
| *ID Dataset* | | | | | | | |
| Imagenet-100 vs cifar-10 | $0.805(L)$ | $< 0.001$ | $0.766(L)$ | $< 0.001$ | $0.547(L)$ | $0.011$ | 16 |
| Imagenet-100 vs cifar-100 | $0.352(M)$ | $0.001$ | $0.320(S)$ | $0.005$ | $0.023(N)$ | $0.211$ | 16 |
| cifar-10 vs cifar-100 | $0.758(L)$ | $< 0.001$ | $0.742(L)$ | $< 0.001$ | $0.531(L)$ | $0.013$ | 16 |
| *Classes_Samples* | | | | | | | |
| c10_s1000 vs c50_s200 | $0.969(L)$ | $< 0.001$ | $1.000(L)$ | $< 0.001$ | $0.453(M)$ | $< 0.001$ | 16 |
| c10_s1000 vs c100_s100 | $1.000(L)$ | $< 0.001$ | $1.000(L)$ | $< 0.001$ | $0.309(S)$ | $< 0.001$ | 16 |
| c10_s1000 vs c100_s200 | $1.000(L)$ | $< 0.001$ | $1.000(L)$ | $< 0.001$ | $0.477(L)$ | $< 0.001$ | 16 |
| c10_s1000 vs c100_s500 | $0.992(L)$ | $< 0.001$ | $1.000(L)$ | $< 0.001$ | $0.707(L)$ | $< 0.001$ | 16 |
| c10_s1000 vs c100_s700 | $0.984(L)$ | $< 0.001$ | $1.000(L)$ | $< 0.001$ | $0.754(L)$ | $< 0.001$ | 16 |
| c50_s200 vs c100_s100 | $0.422(M)$ | $< 0.001$ | $0.469(M)$ | $< 0.001$ | $0.223(S)$ | $< 0.001$ | 16 |
| c50_s200 vs c100_s200 | $0.242(S)$ | $< 0.001$ | $0.352(M)$ | $< 0.001$ | $0.023(N)$ | $0.324$ | 16 |
| c50_s200 vs c100_s500 | $0.391(M)$ | $< 0.001$ | $0.492(L)$ | $< 0.001$ | $0.402(M)$ | $< 0.001$ | 16 |
| c50_s200 vs c100_s700 | $0.367(M)$ | $< 0.001$ | $0.500(L)$ | $< 0.001$ | $0.453(M)$ | $< 0.001$ | 16 |
| c100_s100 vs c100_s200 | $0.219(S)$ | $< 0.001$ | $0.180(S)$ | $0.065$ | $0.250(S)$ | $< 0.001$ | 16 |
| c100_s100 vs c100_s500 | $0.047(N)$ | $0.375$ | $0.016(N)$ | $0.495$ | $0.562(L)$ | $< 0.001$ | 16 |
| c100_s100 vs c100_s700 | $0.062(N)$ | $0.323$ | $0.055(N)$ | $0.980$ | $0.617(L)$ | $< 0.001$ | 16 |
| c100_s200 vs c100_s500 | $0.203(S)$ | $0.004$ | $0.211(S)$ | $0.004$ | $0.383(M)$ | $< 0.001$ | 16 |
| c100_s200 vs c100_s700 | $0.180(S)$ | $0.039$ | $0.250(S)$ | $0.013$ | $0.434(M)$ | $< 0.001$ | 16 |
| c100_s500 vs c100_s700 | $0.008(N)$ | $0.404$ | $0.141(N)$ | $0.065$ | $0.113(N)$ | $0.001$ | 16 |

# F  Classes of ImageNet-100 ID Dataset

We list the 100 classes present in the ID dataset, ImageNet-100 [58]. This list can also be found at: `https://github.com/HobbitLong/CMC/blob/master/imagenet100.txt`

*Rocking chair, pirate, computer keyboard, Rottweiler, Great Dane, tile roof, harmonica, langur, Gila monster, hognose snake, vacuum, Doberman, laptop, gasmask, mixing bowl, robin, throne, chime, bonnet, komondor, jean, moped, tub, rotisserie, African hunting dog, kuvasz, stretcher, garden spider, theater curtain, honeycomb, garter snake, wild boar, pedestal, bassinet, pickup, American lobster, sarong, mousetrap, coyote, hard disc, chocolate sauce, slide rule, wing, cauliflower, American Staffordshire terrier, meerkat, Chihuahua, lorikeet, bannister, tripod, head cabbage, stinkhorn, rock crab, papillon, park bench, reel, toy terrier, obelisk, walking stick, cocktail shaker, standard poodle, cinema, carbonara, red fox, little blue heron, gyromitra, Dutch oven, hare, dung beetle, iron, bottlecap, lampshade, mortarboard, purse, boathouse, ambulance, milk can, Mexican hairless, goose, boxer, gibbon, football helmet, car wheel, Shih-Tzu, Saluki, window screen, English foxhound, American coot, Walker hound, modem, vizsla, green mamba, pineapple, safety pin, borzoi, tabby, fiddler crab, leafhopper, Chesapeake Bay retriever, and ski mask.*

**Is there any semantic class overlap between ID and OOD datasets?** There is no semantic class overlap between ImageNet-100 (ID dataset) and 8 other OOD datasets e.g., CIFAR-10, CIFAR-100, NINCO-64, CUB-200, Aircrafts-100, Oxford Pets-37, Flowers-102, and STL-10.

Only ImageNet-R (consisting of 200 classes) has 19 classes that overlap with ImageNet-100. This is expected and we know that ImageNet-R includes classes from ImageNet-1K dataset but incorporates significant distribution shifts using artistic renditions. The overlapping classes are: *Gasmask, American lobster, Standard poodle, Red fox, Head cabbage, Harmonica, Ambulance, Gibbon, Pineapple, Chihuahua, Tabby, Pirate, Rottweiler, Lorikeet, Boxer, Pickup, Goose, Shih-Tzu, and Meerkat.*

Also, there is no overlap between CIFAR-10 and CIFAR-100 so using one as ID and the other as OOD retains OOD challenges. It is evident that OOD evaluations in all our experiments are substantially robust due to dissimilar classes and significant distribution shifts between ID and OOD datasets.

