# OpenReview forum: "What Variables Affect Out-of-Distribution Generalization in Pretrained Models?"
_NeurIPS.cc/2024/Conference — NeurIPS 2024 poster_

### Official Review · Reviewer_gXLk · 2024-07-12

**Soundness:** 2
**Presentation:** 3
**Contribution:** 2
**Rating:** 4
**Confidence:** 4

**Summary:**

The paper sets out to evaluate empirically the generality (referred to as "universality") of the "tunnel hypothesis", the idea that compression at later layers may hinder OOD performance.

**Strengths:**

Work on OOD learning, network probing and explainability is very relevant to the conference.

**Weaknesses:**

How empirical evaluations may provide any conclusive answer to the question of generality of a hypothesis is unclear to me.

The use of linear probing cannot be taken as the yardstick for this study. The validity of any result is further compromised by the choice of SHAP in the place of a sound knowledge extraction method for the evaluation of the results. SHAP is known to produce unreliable results.

**Questions:**

What does it mean to "disentangle samples and classes"?

What is "diversity of inputs"?

**Limitations:**

Please define linear probe and probe accuracy and how it is used in the paper.

It is very unclear how swapping tunnels might help you answer the question of mitigating forgetting.

---

> ### Author Rebuttal · Authors · 2024-08-05
>
> Thank you for your insightful reviews and feedback. We have carefully considered your concerns. Below, we have responded to your comments.
> # Weaknesses
> **W1. How do empirical evaluations support the generality of a hypothesis?**
> - We addressed this by considering a diverse range of experimental settings, including various deep neural network (DNN) architectures, datasets, and testing conditions. As noted in Appendix A.10, our work involved training 64 different backbones and training/analyzing 10,652 linear probes, resulting in an aggregate compute time of approximately 1,161 hours (48 days). This demonstrates the substantial effort invested in our experimental evaluation. Our experimental results and statistical analyses robustly support our primary findings and hypothesis.
> - Although we did not present theoretical perspectives, we believe our extensive evaluations demonstrated that our revised hypothesis is not just a product of specific conditions but is broadly applicable across various scenarios. While theoretical contributions are undoubtedly important, our work has a profound impact on the field and will inspire future theoretical research. We will mention the limitations regarding theoretical contributions in our paper as follows:
> > Our work empirically examines the tunnel effect and OOD generalization without presenting theoretical insights. Future research may develop theoretical frameworks to explain the revised tunnel effect hypothesis in the context of OOD generalization.
>
> **W2. How effective is linear probing for the OOD study?**
> - The previous work [1] introducing the tunnel effect hypothesis also relied heavily on linear probing for their analyses, further validating its effectiveness in this context.
> - As discussed in Sec. 3.1, linear probing is a widely recognized and standard evaluation technique across various domains, including transfer learning, OOD generalization, and self-supervised learning. We cited 13 papers that extensively use linear probing, although many more could be referenced.
>
> **W3. How reliable is SHAP analysis?**
> - According to NeurIPS'17 paper [2] that introduced SHAP and has been cited $\sim$ 25K times, SHAP is a unified measure of feature importance, with theoretical guarantees that it achieves a unique solution with desired properties.
> - SHAP is widely used in machine learning [2-8] to explain DNNs, providing consistent and reliable interpretations compared to other methods such as SP-LIME, SpRay, GAA, and ACE [7]. Due to SHAP's consistent and superior performance compared to other methods, it is extensively used in the medical field [8].
> - SHAP allows us to control for variables where this careful pairing could not be done. Our SHAP analysis is consistent with our statistical findings using carefully controlled experiments.
>
> **References**
> - [1] Masarczyk et al., The Tunnel Effect: Building Data Representations in Deep Neural Networks, NeurIPS 2023.
> - [2] Scott M. Lundberg et al., A Unified Approach to Interpreting Model Predictions, NeurIPS, 2017
> - [3] Scott M. Lundberg et al., Explainable AI for Trees: From Local Explanations to Global Understanding, https://arxiv.org/abs/1905.04610 (2019)
> - [4] Lundberg, S. et al., An unexpected unity among methods for interpreting model predictions, https://arxiv.org/abs/1611.07478 (2016).
> - [5] Dubois, Yann, et al., "Evaluating self-supervised learning via risk decomposition." In ICML, 2023.
> - [6] S. Mishra et al., Local interpretable model-agnostic explanations for music content analysis., ISMIR, Vol. 53, 2017, pp. 537–543.
> - [7] S., Rabia, et al., "Explaining deep neural networks: A survey on the global interpretation methods." Neurocomputing 513 (2022): 165-180.
> - [8] Lapuschkin S, et al., Unmasking Clever Hans predictors and assessing what machines really learn. Nature communications 10, no. 1 (2019): 1096.
>
> # Questions
> **Q1. What does it mean to "disentangle samples and classes"?**
> If we increase the number of classes, we could also increase the number of samples by just adding in more classes. So, when we increased the classes we kept the total size of the dataset constant. In contrast, [1] increased the classes and samples, so we wanted to disentangle these variables to know whether more classes or more samples mattered more for reducing tunnel effect strength.
>
> **Q2. What is "diversity of inputs"?**
> The diversity of inputs/ dataset refers to factors such as augmentations (within-class diversity), semantic categories (between-class diversity), and resolution, which increase the variability of the training dataset.
>
> # Limitations
> **Please define linear probe and probe accuracy and how it is used in the paper.**
> We discussed the motivation, significance, and process of linear probing in Section 3.1 (motivation and usage), Appendix A.2 (details of linear probing), and A.8 (accuracy). The linear probe training details are discussed in Appendices A.3, A.4, and A.5.
>
> **It is unclear how swapping tunnels might help answer the question of mitigating forgetting.**
> We adopted this experimental design from [1]. Swapping tunnels allows us to isolate the impact of the tunnel itself on mitigating forgetting. For example, let a model be trained on task 1 (extractor $E_1$ and tunnel $T_1$) and then sequentially trained on task 2 (extractor $E_2$ and tunnel $T_2$). When evaluated on task 1, the model's accuracy degrades (forgetting) due to task 2. Here we ask: Does the tunnel $T_1$ contribute to this forgetting? If $T_1$ had no impact, the swapping wouldn’t impact forgetting, meaning $E_1$ with $T_2$ ($E_1+T_2$ without task-specific $T_1$) would achieve similar performance as $E_1+T_1$ and $E_1$ alone would impact forgetting. It turns out $T_1$ significantly impacts forgetting. Likewise, when evaluated on task 2, $E_2+T_1$ achieves lower accuracy than $E_2+T_2$, indicating $T_2$ impacts task 2 performance.
>
> Thank you for your valuable comments & questions. Please let us know if you have further questions.

---

> ### Author Response · Authors · 2024-08-12
>
> Dear Reviewer gXLk,
>
> Thank you for dedicating your time to reviewing our work. Your feedback has been instrumental in enhancing our research.
>
> We understand you may have a busy schedule, but we would greatly appreciate it if you could review our responses to ensure we have fully addressed your concerns. If you have any further questions or suggestions, please do not hesitate to share them. We are committed to addressing any additional points you may have.
>
> Thank you once again for your valuable contribution to our research.

---

### Official Review · Reviewer_WrHJ · 2024-07-12

**Soundness:** 3
**Presentation:** 3
**Contribution:** 3
**Rating:** 7
**Confidence:** 4

**Summary:**

This paper investigates the "tunnel effect" in NNs introduced in a NeurIPS 2023 paper. According to the “tunnel effect” the deeper layers compress representations, limiting OOD generalization. The authors challenge the prior assumption that the tunnel effect is universal. They imply that it's heavily influenced by training data diversity.
They analyze a large number of NNs on various datasets with varying resolutions and introduce three metrics to quantify the tunnel effect's strength. Using a SHAP-based approach, they disentangle the impact of augmentation, number of classes, sample size, resolution, and the architecture.
Their experiments suggest that increasing the diversity of the training data (more classes, augmentations, and higher image resolutions) mitigates the tunnel effect and improves OOD generalization. Depth and overparameterization have the opposite effect.
Based on these findings, they propose a revised tunnel effect hypothesis where its strength is inversely proportional to training data diversity.

**Strengths:**

Originality: This paper is original, in the sense that it challenges existing assumptions on  the universality of the tunnel effect hypothesis, which might have implications on understanding and improving NN’s OOD generalization ability. The authors propose a revised tunnel effect hypothesis which is a novel contribution.

Quality: This paper is scientifically well-executed. The hypothesis is clear and the experiments are extensive (several NN architectures and dataset (configurations), which greatly helps to support their hypothesis. The SHAP-based analysis they performed to disentangle the relative importance of different variables is solid (through rigorous statistical analysis with Wilcoxon signed-rank tests and Cliff's Delta to measure effect sizes) and provides a quantitative measure of the effects.

Clarity: The paper is very well written and easy to follow. Figures and tables are very helpful to understand the key findings.

Significance: The paper’s findings may have significant implications for the “tunnel effect” hypothesis, though I am a bit skeptical of the new insights this brings us in terms of designing algorithms that have better OOD generalization properties.

**Weaknesses:**

Although the paper is a very solid work, the main worry that I have is about the confounding variables in the experiments and the revised hypothesis, which might limit the significance of the contribution:
* For example, Figure 1 claims that the increased resolution mitigates the tunnel effect. But intuitively, if you increase the resolution of the image, you also increase the resolution of the activation maps in the bottleneck layers. Still this doesn’t tell us anything about the architectures per se. Yes, the tunnel effect can be “sidestepped” if you increase the resolution, but then you run into the same issues if you increase the depth.
* OOD accuracy is measured on a number of datasets (NINCO [61], ImageNet-R [62], CIFAR-100 [60], CIFAR-10 [59], Oxford 102 Flowers [63], CUB- 173 200 [64], Aircrafts [65], Oxford-IIIT Pets [66], and STL-10 [67]). However, I wonder what such experiments tell us about the “kind” of OOD that we are testing for. For example, if one tests for compositional or systematic OOD generalization, it becomes very clear what the learnings are, and which particular flaws the models have. I feel that on these benchmarks it is hard to make the similar conclusions.
* Certain findings I find limited in the sense that they are already known, e.g. “Our metrics reveal that increasing the ID class count, higher resolutions, and using augmentations improve OOD generalization”.
* In these figures/results, typically normalized accuracy or performance retained are discussed. However, in ML it is also important to understand the actual accuracy, and these are not easy to find in the paper. Ideally for each figure/results discussion, the reader would be pointed to a table (in the appendix) that contains absolute numbers on the datasets. For example, it is claimed that "overparametrization negatively impacts OOD generalization .. since increasing overparametrization decreases OOD performance **retained**".
However I feel that without the absolute numbers this claim cannot be made. E.g. if both the IID and OOD performance improved by increasing overparametrization, but the IID performance improved more than OOD performance, one cannot claim that OOD performance decreased, quite contrary.
* L281: “Depth. Our SHAP analysis revealed that increasing depth impairs OOD generalization.” - Depending on how you increase depth (e.g. if you have pooling layers), to me this is an obvious consequence of additional compression that you may have in that case.
* The paper claims that their continual learning results contradict [9] and suggest the “tunnel” plays an essential role in mitigating catastrophic forgetting. However, they only replicate the experiment with ResNet-18 on ImageNet-100. To strengthen this claim, the authors could expand the continual learning analysis to include different architectures and datasets.

**Questions:**

1. This citation has the wrong year and NeurIPS number: “Wojciech Masarczyk, Mateusz Ostaszewski, Ehsan Imani, Razvan Pascanu, Piotr Miło ́s, and Tomasz Trzcinski. The tunnel effect: Building data representations in deep neural networks. Advances in Neural Information Processing Systems, 36, 2024.”.
Please cross-check other references too.
2. SHAP papers should be cited the first time it is mentioned in the text.
3. Could you clarify a bit further the reasoning / logic behind the “Metric 3: ID/OOD Alignment”? I find it a bit hard to wrap my head around it in its current form. Specifically why you choose to have a product of the two values and the respective “baseline” values that are subtracted.
4. What do you mean by “more hierarchical features” in L136? Isn’t the tunnel effect reduced because the size of the activation maps is increased, so the bottleneck is decreased?
5. L275: Perhaps a more appropriate wording here would be "7x7 kernel size" instead of "7x7 stem"?

**Limitations:**

The limitations and societal impacts have been addressed.

---

> ### Author Rebuttal · Authors · 2024-08-05
>
> Thank you for your insightful review, comments, and feedback. We have carefully considered your concerns and tried to address your concerns below.
> # Weaknesses
> **W1. Increased resolution mitigates the tunnel effect, but doesn’t increasing DNN depth in that setting bring the tunnel effect back?**
> - We find that resolution has a larger impact on the tunnel effect than depth, but based on our analysis and intuition, a very deep model could still exhibit the tunnel effect. Our work does suggest that for OOD generalization, shallower models may be preferable over deeper ones.
> - In our global rebuttal (Fig. 3), we show that higher resolution maintains a higher rank (evaluated on the ID dataset, following [1]) than low resolution, which corroborates our findings.
>
> **W2. Benchmarks might not reveal compositional or systematic OOD generalization**
> - We conducted a much more comprehensive study than prior work [1] that introduced the tunnel effect. We used a variety of 8 OOD benchmark datasets in each OOD experiment. We focused on evaluating the strength of the tunnel effect and quantifying the relative impact of variables on it. For more fine-grained analysis of compositional or systematic OOD generalization, more crafted datasets might be necessary, but designing such datasets remains outside the scope of our work. Thank you for your insightful comments.
>
> **W3. Some findings reflect things that are already known**
> - While previous research has shown that larger and more diverse datasets help learn invariant features and improve OOD performance, our study uniquely dissects the relative contributions of factors such as resolution, class counts, dataset size, and augmentations toward OOD generalization. So while some of the variables we study are known to impact OOD generalization, the relative contribution of each variable had not been quantified before our study.
>
> **W4. Actual accuracy is necessary besides normalized accuracy**
> - Following previous work [1], we used normalized accuracy to illustrate the tunnel effect. Otherwise, it is very challenging to interpret figures because some datasets are easier or harder, e.g., because they contain more or fewer classes. We reported the actual accuracy in the appendix (Tables 4 and 11) and verified that they are consistent with our findings (see global rebuttal).
> - We understand your concern regarding the ID and OOD accuracy. To address this issue, we introduced another metric ID/OOD alignment which combines ID and OOD accuracy (calculated as the product of ID and OOD accuracy). As shown in Fig. 2(b), our SHAP results with the ID/OOD alignment demonstrate that overparameterization negatively impacts OOD generalization.
> - We have included results based on actual accuracy in the global rebuttal. We see that the actual accuracy trends are consistent with our findings where variables that improve (degrade) OOD generalization also show higher (lower) ID/OOD alignment.
>
> **W5. Increasing depth impairs OOD generalization seems obvious**
> - Previous work [1] has reached a similar conclusion, but they did not quantify the impact. We agree that this conclusion might seem intuitive, but our goal was to quantitatively measure this phenomenon in a broader context. Also, previous studies independently analyzed depth without taking other variables into account that can impact OOD generalization. In contrast, we presented a more comprehensive study, assessing each variable’s relative importance.
> - While increasing depth impairs OOD generalization, we found that this variable's impact on OOD generalization is much smaller than other variables.
>
> **W6. Continual learning experiments could explore more architectures and datasets**
> - Prior work [1] suggested that the tunnel plays a task-agnostic role, but their observations were made in a toy setting with a small dataset. We revisited this to assess the generality of their finding in a more challenging setting with a larger dataset. This exploration was conducted as an auxiliary experiment and is not part of our main results.
>
> # Questions
> 1. **Citation.** We will verify and correct the citations.
> 2. **Citation for SHAP.** We will ensure that SHAP is cited the first time it is mentioned.
> 3. **Reasons behind ID/OOD alignment metric.** The ID/OOD alignment metric combines ID and OOD accuracy to facilitate comparison. Ideally, a model should achieve high ID and OOD accuracy simultaneously (high ID/OOD alignment). This metric potentially alleviates the concern you raised in Weakness #4. We discussed the reasons for our metrics in Appendix A.8.
> 4. **What did the authors mean by more hierarchical features?**
> We hypothesize that higher resolution results in learning more hierarchical features, which reduces the tunnel effect. By "more hierarchical features," we mean that higher resolution enables the network to learn a progression of information—from simple elements like edges and corners to more complex structures and object parts—across multiple stages of the network [2]. Your interpretation is also valid.
> 5. **Stem or kernel.** In Sec 3.2, we referred to the $k\times k$ kernel of the first layer as the $k\times k$ stem. We will make the suggested revisions.
>
> **Reference**
> - [1] Masarczyk et al., The Tunnel Effect: Building Data Representations in Deep Neural Networks, NeurIPS 2023
> - [2] Yann LeCun et al., “Convolutional Networks and Applications in Vision,” ISCAS, IEEE, 2010
>
> Thank you again for your valuable comments, questions, and suggestions. Please let us know if you have further questions.

---

> ### Author Response · Authors · 2024-08-12
>
> Dear Reviewer WrHJ,
>
> Thank you for dedicating your time to reviewing our work. Your feedback has been instrumental in enhancing our research, and we are pleased to hear that you find our work very interesting. We have made every effort to address your remaining concerns.
>
> We understand your schedule may be busy, but we would greatly appreciate it if you could review our responses to ensure we have fully addressed your points. If you have any further questions or suggestions, please do not hesitate to share them. We are committed to addressing any additional points you may have.
>
> Thank you once again for your valuable contribution to our research.

---

> > ### Comment · Reviewer_WrHJ · 2024-08-12
> >
> > I thank the authors for their additional comments and clarifications.
> >
> > I will maintain my score and I still think this is a solid paper.
> >
> > I have also read the reviews of other reviewers and find it somewhat disappointing that they have not yet engaged in a discussion with the authors.

---

> > > ### Author Response · Authors · 2024-08-13
> > >
> > > Dear Reviewer,
> > >
> > > Thank you for your thorough review and valuable feedback. Your insights have been instrumental in enhancing our work. We truly appreciate the time and effort you dedicated to this review.

---

### Official Review · Reviewer_gM3N · 2024-07-12

**Soundness:** 3
**Presentation:** 2
**Contribution:** 2
**Rating:** 5
**Confidence:** 2

**Summary:**

This paper studies how well self-supervised models transfer after self-supervised pretraining.  Using linear probe experiments on top of frozen encoders, and by varying the depth in the model being fed to the linear probe, they study the effect of network depth on ID performance and OOD (transfer learning) performance.  They find that the previously studied "tunnel effect hypothesis" can be diminished by increasing the diversity of pretraining data (which they demonstrate using data augmentation).

**Strengths:**

The paper builds on previous works to clarify their results and provide important additional context.

**Weaknesses:**

Scope:  The results of the paper seem fairly narrowly scoped.  I am not sure how these findings can really inform pretraining in ways that aren't already well understood (e.g. "use more diverse data" is already a fairly well established paradigm).  Thus it seems like the impact of this paper is fairly limited.

Experimental evaluation seems somewhat limited:  While the authors do run a large number of individual training runs, it is not clear to me that they performed a particularly wide experimental study.

**Questions:**

How can we interpret these results in the context of other methods like SSL, where the learning goals seem to be more directly targeting removal of any tunnel effects?

**Limitations:**

Yes.

---

> ### Author Rebuttal · Authors · 2024-08-05
>
> Thank you for your insightful review, comments, and feedback. We have carefully considered your concerns. Below, we have responded to your comments, and we hope these will address your concerns.
>
> # Weaknesses
>
> **W1. The scope of results seems narrow, reflecting known findings**
> - Our work is much more comprehensive and rigorous than the NeurIPS'23 paper [1] that identified the tunnel effect. In contrast to [1], we examined the tunnel effect in much wider settings incorporating 11 datasets, 3 metrics, 8 variables, several DNN architectures (VGG, ResNet, ViT), model scales (shallow, deeper), and various testing conditions. We are the first to quantify each variable's impact on the tunnel effect. Our findings both support the original tunnel effect while finding that it does not always hold and that its strength varies considerably depending on these variables that were ignored in prior work [1].
> - Our results and findings have a significant impact on the out-of-distribution (OOD) generalization and relevant areas of deep learning. While previous works on OOD find that larger and more diverse datasets help learn more invariant features and improve OOD performance (as discussed in Sec. 2.2), our work is _uniquely_ positioned to reveal the _relative_ contribution of each variable e.g., resolution, class, dataset size, and augmentations toward OOD generalization. So while some of the variables we study are known to impact OOD generalization, the relative contribution of each variable had not been quantified before our study.
>
> **W2. Experimental evaluation seems somewhat limited**
> - We conducted the most comprehensive study to date to unravel the intricate relationships between different variables and OOD generalization, _significantly_ expanding upon prior work [1]. For each OOD experiment and analysis, we utilized eight different OOD datasets and assessed various deep neural network architectures, including CNNs and ViTs, across different scales. We are the first to study the tunnel effect in ViTs and widely used self-supervised learning (SSL) models.
> - While additional experiments could provide further insights, we believe the breadth and depth of our experiments do not compromise the quality of our findings. We think our experimental results and statistical analyses robustly support our primary findings and hypothesis, with implications that extend beyond the scope of our specific study.
> - As discussed in Appendix A.10, our work involved training 64 different backbones and training/analyzing 10,652 linear probes, resulting in an aggregate compute time of $\sim$ 1,161 hours (48 days). This demonstrates the substantial effort invested in our experimental evaluation.
>
> **Reference**
> - [1] Masarczyk et al., The Tunnel Effect: Building Data Representations in Deep Neural Networks, NeurIPS 2023
>
> # Questions
>
> **How can these results be interpreted in the context of SSL, where the learning goals target the removal of the tunnel effect?**
>
> - Thank you for your thoughtful question. Earlier work [1] introducing the tunnel effect solely focused on supervised learning (SL) setting. We also mainly focused on the SL setting and briefly studied self-supervised learning (SSL) models since SSL requires significantly more compute than SL. We spent 48 days of compute time on our comprehensive SL study. Conducting a similar study for SSL would require nearly 480 days of compute time ($\sim10\times$ more). As an academic lab, this extensive computation is beyond our current scope.
> - As discussed in Sec. 4.2 and Appendix C.11, our findings indicate that large-scale SSL models do not exhibit the tunnel effect. However, as mentioned in Sec. 5, due to limited compute resources, we were unable to control variables and perform a similar SHAP analysis on a variety of SSL models (at least 4). While our study focused on SL, the methods and evaluation framework we developed could be extended to SSL. In particular, analyzing the impact of different variables—especially augmentations—on the tunnel effect in SSL models could yield valuable insights.
> - Our results suggest that dataset diversity, particularly through augmentation, plays a critical role in mitigating the tunnel effect and enhancing OOD generalization (Fig. 3). This insight hints at the superior OOD generalization abilities of SSL models. Given that SSL methods typically employ more advanced augmentation techniques, expanding our SHAP analysis to disentangle these augmentation policies could help determine whether the OOD generalization capabilities of SSL models are primarily due to their augmentation strategies or their underlying objective functions.
>
> Thank you again for your valuable comments, questions, and suggestions. We hope our responses have addressed your concerns. Please let us know if you have further questions.

---

> ### Author Response · Authors · 2024-08-12
>
> Dear Reviewer gM3N,
>
> Thank you for dedicating your time to reviewing our work. Your feedback has been instrumental in enhancing our research.
>
> We understand you may have a busy schedule, but we would greatly appreciate it if you could review our responses to ensure we have fully addressed your concerns. If you have any further questions or suggestions, please do not hesitate to share them. We are committed to addressing any additional points you may have.
>
> Thank you once again for your valuable contribution to our research.

---

> > ### Comment · Reviewer_gM3N · 2024-08-13
> > **Response to the authors**
> >
> > I have read the rebuttal from the authors.  I think their response reduces my concerns slightly and will therefore raise my score correspondingly.

---

> > > ### Author Response · Authors · 2024-08-13
> > >
> > > Dear Reviewer,
> > >
> > > We sincerely thank you for your constructive feedback and for reconsidering the score.
> > >
> > > Regarding the concern about scope, we would like to emphasize that our work introduces new perspectives on the tunnel effect, providing researchers and practitioners with insights to _systematically_ design interventions that prevent tunnel formation and enhance OOD transferability or downstream tasks. We firmly believe that our work offers a meaningful contribution to the field and has the potential to inspire future research.
> > >
> > > If you have any further concerns or comments, we would be more than happy to address them.

---

### Official Review · Reviewer_tyj4 · 2024-07-15

**Soundness:** 2
**Presentation:** 2
**Contribution:** 2
**Rating:** 4
**Confidence:** 4

**Summary:**

This paper studies the factors influencing out-of-distribution (OOD) generalization of pre-trained DNN embeddings through the lens of the tunnel effect hypothesis, which suggests deeper DNN layers compress representations and hinder OOD performance.

**Strengths:**

- The paper includes a sufficient amount of experiments, offering detailed and substantial content.

- The main purpose of the paper is clear and easy to understand.

**Weaknesses:**

- The technical contribution is limited. The experiments on OOD generalization discussed in the paper are rather trivial and do not yield sufficiently insightful conclusions.

- The paper is poorly written. As an evaluation-focused study, it fails to effectively organize the relationship between the arguments and experiments, making it rather disorganized.

- The paper does not provide significant theoretical contributions, focusing primarily on empirical results without delving into the underlying theoretical frameworks or offering new theoretical insights.

**Questions:**

please refer to the weakness.

**Limitations:**

yes

---

> ### Author Rebuttal · Authors · 2024-08-05
>
> Thank you for your thorough reviews and insightful feedback. We have carefully considered your concerns and tried to address them. Below, we have provided detailed responses to each review separately.
>
> # Weaknesses
>
> **W1. Limited Technical Contributions**
>
> - We believe our work presents substantial scientific contributions with valuable insights. The main contribution of our work lies in conceptualizing and offering a cohesive perspective on how representations form within deep neural networks (DNNs).
> - The tunnel effect, a recently discovered phenomenon in DNNs, enhances our understanding of out-of-distribution (OOD) generalization. Although initially identified in prior work [1], our research revises and strengthens the hypothesis through in-depth analysis, placing it within a broader context.  Our work replicates their findings for small datasets but finds that the strength of the tunnel effect greatly diminishes due to augmentations, resolution, and number of classes.
> - To the best of our knowledge, we are the first to conduct a comprehensive study to disentangle and precisely measure the impact of different variables on OOD generalization through the lens of the tunnel effect hypothesis.
> ## Our key scientific contributions include:
> 1. **Novel Evaluation Framework.** We introduced a novel SHAP-based evaluation framework that uses multiple metrics to reveal the intricate interactions among variables and their influence on OOD generalization. Our framework enables a more nuanced assessment and comparison of various models' OOD capabilities.
> 2. **Broader Exploration of the Tunnel Effect Hypothesis.** We examined the tunnel effect across different settings, including supervised and self-supervised learning, various architectures, and datasets, and revised the tunnel effect hypothesis to align it with broader contexts.
> 3. **Insightful Findings.** We revealed how different variables impact the tunnel effect individually and collectively, offering essential insights for designing architectures and training pipelines to enhance OOD generalization. Our experiments yielded significant insights, such as how the tunnel effect varies across the same architecture and dataset (Figs. 1-4).
> 4. **Continual Learning Perspectives.** Our findings demonstrated the tunnel effect's role in mitigating catastrophic forgetting in continual learning (CL) scenarios. In contrast to previous work [1], we find that the tunnel plays a task-specific role and significantly impacts forgetting in CL. Our findings provide invaluable perspectives on mitigating catastrophic forgetting for the CL research community.
> 5. **Explanation for Scaling Issues.** We provided the first evidence that the inherent characteristics of toy datasets exacerbate the tunnel effect, hindering the learning of reusable features and limiting generalization to larger, real-world datasets. Our work questions the generalizability of results obtained from toy datasets and underscores the importance of conducting diverse tests in deep learning research to ensure algorithm scalability.
>
> **W2. Poor Writing and Organization**
> - We carefully crafted our paper to clearly communicate our research outcomes. After numerous revisions and months of refinement, we believe the paper is well-organized, making it easier to understand the motivation, methodology, and conclusions of each experiment and analysis. However, we will address the reviewers' concerns and further revise the paper to enhance the clarity and quality of our presentation.
>
> **W3. Lack of Theoretical Contributions**
>
> - Our work builds on a prior NeurIPS paper [1] that empirically examined the tunnel effect phenomenon. To align the tunnel effect with broader contexts, we conducted a much more comprehensive study than [1]. We carefully measured tunnel effect strength and controlled for all relevant variables.
> - While theoretical contributions are undoubtedly important, we believe theory requires empirical science to come first. Our empirical findings significantly contribute to the field and will inspire future theoretical research. We will mention the limitations regarding theoretical contributions in our paper as follows:
>
> > Our work empirically examines the tunnel effect and OOD generalization without presenting theoretical insights. Future research may develop theoretical frameworks to explain the revised tunnel effect in the context of OOD generalization.
>
> Thank you again for your valuable comments and suggestions. We hope our responses have addressed your concerns. Please let us know if you have further questions.
>
> **Reference**
> - [1] Masarczyk et al., The Tunnel Effect: Building Data Representations in Deep Neural Networks, NeurIPS 2023.

---

> > ### Comment · Reviewer_tyj4 · 2024-08-14
> >
> > Thanks for your responses. Unfortunately, the paper's technical and theoretical contributions are insufficient, so I will maintain my original score, which is below the acceptance threshold.

---

> ### Author Response · Authors · 2024-08-12
>
> Dear Reviewer tyj4,
>
> We sincerely appreciate your time in reviewing our work. Your feedback has been invaluable in improving our research.
>
> We understand you may have a busy schedule, but we would greatly appreciate it if you could review our responses to ensure we have fully addressed your concerns.  If you have any further questions or suggestions, please feel free to share them. We are committed to addressing any additional points you may have.
>
> Thank you again for your valuable contribution to our research.

---

### Author Rebuttal · Authors · 2024-08-06

We thank the reviewers for their constructive feedback, valuable insights, and thoughtful questions. We have carefully
considered all comments and provided detailed responses to each review separately. We have revised our paper as we should.
We hope our responses have addressed the reviewers' concerns. Please take a look at the attached PDF for additional analyses.

## Contribution and Impact
Below we have mentioned some points to highlight our work's key scientific contribution and impact.
- Our main contribution is a _significant_ expansion of the findings from the original tunnel effect paper [1], which did not measure many of the variables we investigated. We found that the formation of tunnels greatly influences out-of-distribution (OOD) generalization, and by quantifying the impact of each variable through our SHAP analysis—a method widely used in other areas of deep learning but largely overlooked in vision—we provide valuable insights into how OOD generalization can be further improved.

- In addition to OOD generalization, many algorithms, such as OOD detection methods, may benefit from understanding the tunnel effect. While our primary focus is on measuring the impact of each variable, we also highlight an important issue: the representations learned by deep neural networks for vision tasks can have very different properties by using augmentations, higher resolution, or more classes. By showcasing this impact, our paper contributes to addressing machine learning's replicability crisis, where an algorithm may fail to perform consistently across different datasets. Our work helps explain why such discrepancies occur in vision tasks.

**Reference.**
[1] Masarczyk et al., "The Tunnel Effect: Building Data Representations in Deep Neural Networks", NeurIPS 2023

We appreciate your feedback and the opportunity to provide clarifications. Please let us know if you have further questions or concerns. We will try our best to address your concerns.

---

### Decision · Program_Chairs · 2024-09-25

**Decision:**

Accept (poster)

**Comment:**

This paper provides a thorough investigation of a recently observed phenomenon in deep learning called the "tunnel effect." The hypothesis states that initial layers create linearly-separable representations, while subsequent layers (referred to as the "tunnel") compress these representations with minimal impact on overall performance. This paper verifies the existence of the tunnel effect and demonstrates that it can be mitigated with richer data. Additionally, the authors verify that representations created by compression layers can negatively impact out-of-distribution (OOD) generalization performance. These claims are supported across multiple architectures, datasets, and other learning settings (including self-supervised learning). Compared to the original paper that discovered this phenomenon, this submission offers a much broader empirical analysis.

The majority of reviewers find the paper borderline, with the exception of reviewer gM3N, who rates the paper as "accept." Concerns raised by reviewers with borderline ratings include a lack of theoretical support and unsurprising conclusions. In contrast, reviewer gM3N, who provided the most thorough review of this submission, believes the paper makes an important contribution with potential implications for understanding and improving OOD generalization ability, as it challenges existing assumptions about the universality of the tunnel effect hypothesis.

The overall rating for the paper is borderline accept. After reading all the reviews and responses, I tend to agree with reviewer gM3N. Not every paper needs a theoretical contribution, especially when it is thorough from an empirical standpoint, and I think the authors have done a reasonable job in their empirical investigation. I also believe that verifying the existence of the tunnel effect while also showing that it is not a universal effect, and doing so across different learning settings (as done in the paper), is a non-trivial contribution. I therefore recommend acceptance.